# Implications of the ongoing rock uplift in NW Himalayan interiors

Saptarshi Dey[1], Rasmus Thiede[2], Arindam Biswas[3], Naveen Chauhan[4], Pritha Chakravarti[1], and Vikrant Jain[1]

[1]*Earth Science Discipline, IIT Gandhinagar, Gandhinagar-382355, India.*

[2]*Institute of Geosciences, Christian Albrechts University of Kiel, Kiel-24118, Germany.*

[3] *Department of Applied Geology, IIT-ISM Dhanbad, Jharkhand-826004, India.*

[4] *Atomic Molecular and Optical physics Division, Physical Research Laboratory, Ahmedabad.*

Corresponding author

Saptarshi Dey

saptarshi.dey@iitgn.ac.in

**Abstract**

The Lesser Himalayan duplex exposed in the Kishtwar Window (KW) of the Kashmir Himalaya exhibits rapid rock uplift and exhumation (~3 mm/yr) at least since the Late Miocene. However, it has remained unclear if it is still actively-deforming. Here, we combine new field observations, morphometric and structural analyses with dating of geomorphic markers to discuss the spatial pattern of deformation across the window. We found two steep stream segments, one at the core and the other along the western margin of the KW, which may possibly be linked to crustal ramps on the MHT. Longitudinal fluvial profiles document gradients changes across the entire length of the window, and high gradient changes in the core of the window. High bedrock incision rates (> 3 mm/yr) are deduced from dated strath terraces along deeply-

incised Chenab River valley lying above the potential ramp along the western margin of the KW. In contrast, farther downstream on the hanging wall of the MCT, fluvial bedrock incision rates are lower (< 0.8 mm/yr). Bedrock incision rates largely correlate with previously-published thermochronologic data. The obtained results can be partially explained by existence of multiple crustal ramps which could result into differential uplift due to translation on the basal decollement. Or, similar rock uplift can also be caused by out-of-sequence faulting at the core and along the western margin of the window. In summary, our study highlights a structural and tectonic control on landscape evolution over millennial timescales.

**Keywords**

Steepness index; knickzone, rock strength; bedrock incision; Main Himalayan Thrust.

## 1. Introduction

Protracted convergence between the Indian and the Eurasian plate resulted into the growth and evolution of the Himalayan orogen and temporal in-sequence formation of the Southern Tibetan Detachment System (STDS), the Main Central Thrust (MCT), the Main Boundary Thrust (MBT) and the Himalayan Frontal Thrust (HFT) towards the south (e.g., Yin and Harrison, 2000; DiPietro and Pogue, 2004) (Supplementary Fig.B1). HFT defines the southern termination of the Himalayan orogenic wedge and separates the orogen from the undeformed foreland basin known as the Indo-Gangetic Plains. Seismic reflection profiles reveal that all these fault-zones emerge from a low-angle basal decollement, the Main Himalayan Thrust (MHT) forming the base of the Himalayan orogenic wedge (e.g., Ni and Barazangi, 1984;

Nabelek et al., 2009; Avouac et al., 2016), established in the late Miocene (Vannay et al., 2004).
Existence of MHT has further been elaborated in Himalayan cross-sections (e.g., Powers et al.,
1998; Decelles et al., 2001; Webb et al., 2011; Gavillot et al., 2018).

Lave and Avouac (2000) studied the late Pleistocene-Holocene shortening history of the

Central Nepal Himalaya where they showed the Holocene shortening is accommodated only
across the HFT. However, a large body of literature in the eastern, central and western Himalaya
favored that majority of the late Pleistocene-Holocene shortening is rather partitioned throughout
the Sub-Himalayan domain (morphotectonic segment in between the MBT and the MFT) and not
solely accommodated by the HFT (e.g., Wesnousky et al., 1999; Burgess et al., 2012; Thakur et
al., 2014; Mukherjee, 2015; Vassalo et al., 2015; Dey et al., 2016; Dey et al., 2018). The
statement above implies that the northerly thrusts, i.e., the MBT and the brittle faults exposed in
the vicinity of the southern margin of the Higher Himalaya, are considered inactive over
millennial timescales. However, in recent years, several studies which focused on the low-
Temperature thermochronologic data and thermal modeling of the interiors of the NW Himalaya
have raised questions on the statement above. The recent studies suggested that 1-3 mm/yr out of
the total Quaternary shortening has been accommodated in the north of the MBT as out-of-
sequence deformation (Thiede et al., 2004; Deeken et al., 2011; Thiede et al., 2017) or in form of
growth of the Lesser Himalayan Duplex (Gavillot et al., 2018) (Supplementary Fig. B2). For
faults within the hinterland of the Central Himalaya, the out-of-sequence deformation has been
explained by two end-member models. One of them favored the reactivation of the MCT (Wobus
et al., 2003), while the other tried to explain all changes along the southern margin of the Higher
Himalaya driven by enhanced rock uplift over a major ramp on the MHT (Bollinger et al., 2006;
Herman et al., 2010; Robert et al., 2009). Landscape evolution models, structural analysis and
thermochronologic data from the interior of the Himalaya favor that the Lesser Himalaya has
formed a duplex at the base of the southern Himalayan front by sustained internal deformation
since late Miocene (Decelles et al., 2001; Mitra et al., 2010; Robinson and Martin, 2014; Gavillot
et al., 2016). The growth of the duplex resulted into the uplift of the Higher Himalaya forming
the major orographic barrier of the orogen. The Kishtwar Window (KW) in the NW Himalaya
represents the northwestern termination of the Lesser Himalayan Duplex (LHD). While most of
the published cross-sections of the Himalayan orogen today recognize the duplex structures
within the Lesser Himalaya (Webb et al., 2011; Mitra et al., 2010; DeCelles et al., 2001; Gavillot
et al., 2018), little or no data are available on how the deformation is spatially as well as
temporally distributed and most importantly, whether a duplex is active over millennial
timescales.
The low-temperature thermochron study by Kumar et al., (1995) portrayed the first
orogen-perpendicular sampling traverse extending from the Kishtwar tectonic Window over the
Zanskar Range. More recent studies link the evolution of the KW to the growth of the Lesser
Himalayan Duplex structure (Gavillot et al., 2018), surrounded by the Miocene MCT shear zone
along the base of the High Himalayan Crystalline, locally named as the Kishtwar Thrust (KT)
(Ul Haq et al., 2019). Thermochronological constraints suggest higher rates of exhumation
within the window (3.2-3.6 mm/yr) with respect to the surroundings (~0.2 mm/yr) (Gavillot et
al., 2018), corroborating well with similar thermochron-based findings from the of the Kullu-
Rampur window along the Beas (Stübner et al., 2018) and Sutlej valley (Jain et al., 2000;
Vannay et al., 2004; Thiede et al., 2004) over the Quaternary timescale. No evidence exists
whether the hinterland of the Kashmir Himalaya is tectonically-active over intermediate
timescales. Therefore, to understand the $10^3$-$10^4$-year timescale neotectonic evolution, we

combined geological field evidences, chronologically-constrained geomorphic markers and morphometric analysis of potential study areas, such as the KW. The detailed structural information of the window and the surroundings, previously-published thermochron data, accessibility, well-preserved sediment archive, and recognizable geomorphic markers across the Kishtwar Window makes it a potent location for our study.

In this study, we focus on the following long-standing questions on Himalayan neotectonic evolution, which are-

1. Is there any ongoing neotectonic deformation in the interiors of the Kashmir Himalaya?

2. Can we determine sub-surface structural variations and existence of surface-breaking faults by analyzing terrain morphology?

3. Can we obtain new constraints on deformation over geomorphic timescales? Do millennial-scale fluvial incision rates support long-term exhumation rates?

To address these questions, we adopted a combination of methods such as morphometric analysis using high-resolution digital elevation models, field observation on rock type, structural variations as well as rock strength data and, analysis of satellite images to assess the spatial distribution of the late Quaternary deformation of the KW and surroundings (Fig.1). We aimed to evaluate the role of active tectonics and geometric variations in the basal decollement in shaping the topography (Fig.1). We used basinwide steepness indices and specific stream power as a proxy of fluvial incision. And, lastly but most importantly, we calculated the fluvial bedrock incision rates by using depositional ages of aggraded sediments along Chenab River. In this study, we show that the regional distribution of topographic growth is concentrated in the core of

the window and along the western margin of the window. Our new estimates on the bedrock
incision rate agree with Quaternary exhumation rates from the KW, which could mean consistent
active growth of the Kishtwar Window over million-year to millennial timescales. Although the
observed topographic and morphometric pattern indicate a structural/tectonic control on
topographic evolution, with the available data we are not able to resolve whether it is caused by
passive translation on the MHT or by active surface-breaking faulting within the duplex.

**2.   Geological background**
Regionally balanced cross-sections (DiPietro and Pogue, 2004; Searle et al., 2007;
Gavillot et al., 2018) suggest that the Himalayan wedge is bounded at the base by décollement,
named the MHT and all regionally-extensive surface-breaking thrust systems are rooted to it.
The orogenic growth of the Himalaya resulted into an overall in-sequence development of the
orogen-scale fault systems which broadly define the morphotectonic sectors of the orogen (Fig.
1b). Notable among those sectors, the Higher Himalaya is bordered by the MCT in the south and
is comprised of high-grade metasediments, Higher Himalayan Crystalline Sequence (HHCS) and
Ordovician granite intrusives (Fuchs, 1981; Steck, 2003; DiPietro and Pogue, 2004; Gavillot et
al., 2018). The Low-grade metasediments (quartzites, phyllites, schists, slates) of the Proterozoic
Lesser Himalayan sequence are exposed between the MCT in the north and MBT in the south.
The Lesser Himalayan domain is narrow (4-15 km) in the NW Himalaya except where it is
exposed in the form of tectonic windows (Kishtwar window, Kullu-Rampur window etc.) in the
western Himalaya (Steck, 2003). The Sub-Himalayan fold-and-thrust belt lying to the south of
the MBT is tectonically the most active sector since the late Quaternary (Gavillot, 2014; Vassallo
et al., 2015; Gavillot et al., 2018).
Near the southwest corner of our study area, Proterozoic low-grade Lesser Himalayan
metasediments are thrust over the Tertiary Sub-Himalayan sediments along the MBT (Wadia,
1934; Thakur, 1992). Near the Chenab region in the Kashmir Himalaya, Apatite U-Th/He ages
suggest that cooling and exhumation related to faulting along the MBT thrust sheet initiated
before ~5 ± 3 Myr (Gavillot et al., 2018). Geomorphic data obtained across the MBT in Kashmir
Himalaya suggest that MBT has not been reactivated for the last 14-17 kyr (Vassallo et al.,
2015). In the Kashmir Himalaya, the Lesser Himalayan sequence (LHS) exposed between the
MBT and the MCT is characterized by a < 10 km-wide zone of sheared schists, slates, quartzites,
phyllites and Proterozoic intrusive granite bodies (Bhatia and Bhatia, 1973; Thakur, 1992; Steck,
2003). The LHS is bounded by the MCT shear zone in the hanging wall. The MCT hanging wall
forms highly deformed nappe exposing lower and higher Haimantas, which are related to the
Higher Himalayan Crystalline Sequence (HHCS) (Bhatia and Bhatia, 1973; Thakur, 1992; Yin
and Harrison, 2000; Searle et al., 2007; Gavillot et al., 2018). Nearly 40 km NE of the frontal
MCT shear zone, MCT fault zone is re-exposed as a klippe in the vicinity of KW is called the
Kishtwar Thrust (KT) (Ul Haq et al., 2019) (fig. 1). Within the KW, Lesser Himalayan
quartzites, low-grade mica schists and phyllites along with the granite intrusives are exposed
(Fuchs, 1975; Steck, 2003; DiPietro and Pogue, 2004; Yin, 2006; Gavillot et al., 2018).

## 2.1. Structural architecture of the Kishtwar Window

The sub-surface structural formation beneath the KW is not well-constrained. A recent
study by Gavillot et al., (2018) proposes that the KW exposes a stack of LHS nappes in form of
the commonly-known Lesser Himalayan Duplex (LH duplex), characteristic of the central
Himalaya (Decelles et al., 2001). They also propose the existence of two mid-crustal ramps
beneath the KW, viz., MCR-1 and MCR-2 (fig. 1b). Based on thermochronological constraints
from Kumar et al., (1995), Gavillot et al. (2018) proposed that the core of the window is
exhumed with rates 3.2-3.6 mm/yr during the Quaternary, at a higher rate when compared to the
surroundings (~0.2-0.4 mm/yr). However, earlier studies by Fuchs (1975) and Frank et al.,
(1995) provide different insights to the formation of the KW. Fuchs (1975) proposed the
existence of two nappes- a. the Chail Nappe and b. the Lower Crystalline Nappe. The Lower
Crystalline nappe is partially or completely included in the MCT (KT) shear zone and the Chail
nappe encompasses the core of the window (Stephenson et al., 2000). According to these studies,
the Chail nappe has been internally deformed by crustal buckling, tight isoclinal folding causing
repetition and thickening of the LH crust.

The Higher Himalayan sequence dips steeply away from the duplex (~65° towards west)

(Fig.1, 2a). The frontal horses of the LH duplex expose internally-folded greenschist facies
rocks. Although at the western margin of the duplex, the quartzites stand sub-vertically (Fig.2c),
the general dip amount reduces as we move from west to east for the next ~10-15 km up to the
core of the KW. Near the core of the KW, we observed highly-deformed (folded and multiply-
fractured) quartzite at the core of the KW (Fig.2d, 2e). We also observed deformed quartz veins
of at least two generations, as well as macroscopic white mica. Here, the Chenab River is also
very steep and narrow; the rock units are also steeply-dipping towards the east (~55-65°) and are
nearly isoclinal and strongly deformed at places (Fig.2f). Towards the eastern edge of the
window, however, the quartzites dip much gently towards the east (~20-30°) (Fig.1b), and much
lesser folding and faulting have been recognized in the field (Fig.2g).
**2.2.Valley morphology**

The broad, 'U-shaped' valley profile near the town of Padder at the eastern margin of the

KW is in contrast with the interior of the window (Fig.3a). At the core of the KW, the Chenab
River maintains a narrow channel width and a steep gradient (Fig.3b). The E-W traverse of the
Chenab River through the KW is devoid of any significant sediment storage. However, along the
N-S traverse parallel to the western margin of the KW, beneath the Kishtwar surface, ~150-170m
thick sedimentary deposits are transiently-stored over the steeply-dipping Higher Himalayan
bedrock (Fig.3c). The height of the Kishtwar surface from the Chenab River is ~450m, which
means ~280m of bedrock incision by the River since the formation of the Kishtwar surface.
Along the N-S traverse of the River, epigenetic gorges are formed as a result of the damming of
paleo-channel by the hillslope debris flow, followed by the establishment of a newer channel
path (Ouimet et al., 2008; Kothyari and Juyal, 2013). One example of such epigenetic gorge
formation near the town of Drabshalla is shown in Fig.3d. Downstream from the town of
Drabshalla, the River maintains narrow channel width (< 25 m) and flows through a gorge
having sub-vertical valley-walls (Fig.3e). The tributaries originating from the Higher Himalayan
domain form one major knickpoint close to the confluence with the trunk stream (Fig.3f). We
have identified at least three strath surface levels above the present-day river channel, viz., T1
(280±5 m), T2 (170-175 m) and T3 (~120±5 m), respectively (Fig.3g). The first study on
sediment aggradation in the middle Chenab valley (transect from Kishtwar to Doda town) was
published by Norin (1926). He argued the sediment aggradation in and around the Kishtwar town
is largely contributed by fluvioglacial sediments and the U-shaped valley morphology is a
marker of past glacial occupancy. In general, we agree with the findings of Norin (1926) and Ul
Haq et al., (2019) as we observe ~100m thick late Pleistocene fluvioglacial sediment cover
unconformably overlying the Higher Himalayan bedrock, most likely to be paleo-strath surface
(Fig.4b). At the same time, we do not agree with the interpretation of surface-breaking faults
near Kishtwar town by Ul Haq et al. (2019). We inspected the proposed fault locations in detail
and didn't observe any evidence of large-scale fault movement, including offset, broken and
rotated clasts, fault gouges etc. on the proposed fault planes. There is only one evidence of a
deformed sand layer which shows tilting and offset (<1 m). Therefore, we may conclude that we
found no strong evidence of any large-scale surface-breaking faults. The fluvioglacial sediments
included alternate layers of pebble conglomerate and coarse-medium sand (Fig.4c). The pebbles
are moderately rounded and polished suggesting significant fluvial transport. Our field
observations suggest that the fluvioglacial sediments have been succeeded by a significant
volume of hillslope debris flow and paleo-landslide deposit (Fig.4c). The thickness of the debris-
flow deposits is variable. The hillslope debris units and landslide deposits contain mostly
massive, highly-angular, poorly-sorted quartzite clasts from the steep western margin of the KW.
The hillslope debris units also contain a few fine-grain sediment layers trapped in between two
coarse-grained debris layers (Fig.4e). The town of Kishtwar is situated on this debris flow
deposit.

### 3.  Methods of morphometric analysis and field data collection

**3.1.Morphometry**
For conducting the morphometric analysis, we have used 12.5m ALOS-PALSAR DEM
data (high resolution terrain-corrected) (Fig.5a). This DEM data has lesser issues with artifacts
and noises than 30m SRTM data, which fails to capture the drainage network properly in areas
populated by narrow channel gorges. Topographic relief has been calculated using a 4km moving
window (Fig.5b) and the rainfall distribution pattern has been adapted from 12-year averaged
annual rainfall data (TRMM data: Bookhagen and Burbank, 2006) (Fig.5c).

### 3.1.1. Drainage network extraction

The drainage network and the longitudinal stream profiles were extracted using the Topographic Analysis Kit toolbox (Forte and Whipple, 2019). An equivalent of 10-pixel smoothing of the raw DEM data has been applied to remove noises from the DEM. The longitudinal stream profile of the Chenab trunk stream was processed with the Topotoolbox 'Knickpointfinder' tool (Schwanghart and Scherler, 2014). Several jumps/ kinks in the longitudinal profile are seen and those are marked as knickpoints (Fig.6). A 30m tolerance threshold was applied to extract only the major knickpoints.

### 3.1.2. Basinwide normalized steepness indices

Global observations across a broad spectrum of tectonic and climatic regimes have revealed a power-law scaling between the local river gradient and upstream contributing area:

$$S = k_s . A^{-\theta} \qquad (1)$$

where S is the stream gradient (m/m), $k_s$ is the steepness index ($m^{2\theta}$), A is the upstream drainage area ($m^2$), and $\theta$ is the concavity index (Flint, 1974; Whipple and Tucker, 1999). Normalized steepness-index values ($k_{sn}$) are steepness indices calculated using a reference concavity value ($\theta_{ref}$), which is useful to compare steepness-indices of different river systems (Wobus et al., 2006). We extracted the $k_{sn}$ values in the study area using the ArcGIS and MATLAB-supported Topographic Analysis Toolkit (Forte and Whipple, 2019) following the procedure of Wobus et al. (2006). We performed an automated $k_{sn}$ extraction using a critical area of $10^6$ $m^2$ for assigning the channel head, a smoothing window of 500 m, a $\theta_{ref}$ of 0.45, and an auto-$k_{sn}$ window of 250 m for calculating $k_{sn}$ values. The slope-breaks, known as the knickpoints (sometimes referred to as knickzones if it is manifested by a series of rapids instead of a single sharp break in profile), were allocated by comparing the change of slope along the distance-

elevation plot (Fig.6, 7a). Threshold 'dz' value (projected stream offset across a knickpoint) for
this study is 30m. Basinwide mean $k_{sn}$ values are plotted using a 1000 km$^2$ threshold catchment
area (Fig. 5d).
Identification of the knickpoints/ knickzones and their relationship with the rock-types as
well as with existing structures are necessary to understand the causal mechanism of the
respective knickpoints/ knickzones. Knickpoints/(zones) can be generated by lithological,
tectonic and structural control. Lithological knickpoints are stationary and anchored at the
transition from the soft-to-hard substrate. The tectonic knickpoints originate at the active tectonic
boundary and migrate upstream with time. Structural variations, such as thrust fault ramp-flat
geometry may cause a quasistatic knickpoint at the transition of the flat-to-ramp of the fault. In
such cases, the ramp segment is characterized by higher steepness than the flat segment and at
times the ramp may be characterized by a sequence of rapids, forming a wide knickzone, instead
of a single knickpoint.
### 3.1.3.  Channel Width
Channel width is a parameter of assessment of lateral erosion/incision through bedrocks
of equivalent strength (Turowski, 2009). The channel width of the Chenab trunk stream from just
downstream of the MBT up to the eastern margin of the KW was derived by manual selection
and digitization of the channel banks using the Google Earth Digital Globe imagery
(http://www.digitalglobe.com/) of minimum 3.2 m spatial resolution. We used the shortest
distance between the two banks as the channel width. We rejected areas having unparallel
channel-banks as that would bias the result. We used a 50 m step between two consecutive points
for channel width determination. Twenty point-averaged channel width data along with elevation
of the riverbed is shown in Fig.7b.
*3.1.4. Specific stream power (SSP) calculation*

Specific stream power has often been used as a proxy of fluvial incision or differential

uplift along the channel (Royden and Perron, 2013; Whipple and Tucker, 1999). Areas of higher
uplift/incision are characterized by transient increase in the specific stream power. Channel slope
and channel width data were used to analyse the corresponding changes in the specific stream
power (SSP) from upstream of the gorge area to the gorge reaches (Bagnold, 1966). The SSP ($\omega$)
was estimated using the following equation –
$$\omega = \gamma.Q.s/w \qquad\qquad \text{(Eq. 1)}$$

Where, $\gamma$ - unit weight of water, Q – water discharge, s – energy slope considered

equivalent to the channel slope; w – channel width. SSP data from selected stretches are shown
in Table 1.

**3.2. Field data collection**
*3.2.1. Structural data*

We measured the strike and dip of the foliations and bedding planes of the Lesser and

Higher Himalayan rocks using the Freiberg clinometer compass. At least five measurements are
taken at every location and the average of them has been reported in Fig. 8a. Field photos in the
Fig.2 support observed variations in the structural styles.
*3.2.2. Rock strength data*

Recording rock strength data in the field is important to understand the role of variable

rock-type and rock-strength in changes in morphology. It provides us important insights on the
genesis of knickpoints whether they are lithologically-controlled or not. It also helps to
understand the variations in channel steepness across rocks of similar lithological strength. We
systematically measured the rock strength of the main geologic units using a hand-held rebound
hammer. Repeated measurements (8-10 measurements at each of the 75 locations throughout the
study area) were conducted to measure the variability of rock-strength within the main lithologic
units (Fig. 7e). All the measurements were taken perpendicular to the bedding/ foliation plane,
and, no measurements are from wet surfaces or surfaces showing fractures. Each reading was
taken at least 0.5m apart from the previous one. To our benefit, most of the road-cut sections had
bedrock-exposures. Except restricted locations, e.g., dam-sites and military bases and outposts,
we were able to cover rest of the study area. To add to this, data taken from Higher Himalayan
intrusives close to the western margin of the KT are positively-biased as it represents readings
only from the leucosomatic layers. Our data from individual sites are smaller in number than
what is preferred for checking the statistical robustness of Schmidt hammer data (Niedzielski et
al., 2009). Therefore, we combined the data from all sites representing similar lithology and
portrayed the mean ±standard deviation for the same. Field data on rock strength measurement
has been provided in Supplementary Table C1.
**3.3.Luminescence dating of transiently-stored sediments in and around Kishtwar**

Luminescence dating of Quaternary sediments is a globally accepted method for

constraining the timing of deposition of sediments across different depositional environments,
viz., Aeolian (Juyal et al., 2010), fluvial (Olley et al., 1998; Cunningham and Wallinga, 2012)
and glacial origin (Owen et al., 2002; Pant et al., 2006). In this study, we used luminescence
dating techniques to constrain depositional ages of several fluvioglacial and fluvial sand layers
exposed near the western margin of the KW and further downstream. Although there exists a few
persistent problems in luminescence dating of the Himalayan sediments (including poor
sensitivity of quartz and numerous cases of heterogeneous bleaching of the luminescence signal),
studies over the past couple of decades have also provided a good control on Himalayan
sedimentary chronology by using luminescence dating with quartz (Optically stimulated
luminescence, OSL) and feldspar (Infra-red stimulated luminescence, IRSL).
Samples K-07, K-08 and K-09 were collected from the medium-coarse sand beds of
fluvioglacial origin and have been dated with IRSL technique (Preusser, 2003). Standard IR-
protocol was used because the OSL signal was saturated and postIR-IR was showing instances of
heterogeneous bleaching. Samples K-02 and K-11 were taken from the fine sand-silt layers lying
above the debris-flow deposits and have been treated for OSL dating using double-SAR (single
aliquot regenerative) protocol (Roberts, 2007). Double-SAR protocol was used to surpass the
luminescence signal from tiny feldspar inclusions within individual quartz grains. Samples K-16
and K-17 taken above the T3 strath level, as well as the sample K-18, taken from above the T1
strath level were treated/ measured following the OSL double-SAR protocol. Samples K-01 and
K-06 taken above the bedrock strath near the town of Doda were also measured following OSL
double-SAR protocol. The aliquots were considered for equivalent dose (ED) estimation only if:
(i) recycling ratio was within $1\pm0.1$, (ii) ED error was less than 20%, (iii) test dose error was less
than 10%, and (iv) recuperation was below 5% of the natural. Fading correction of the IRSL
samples K-07 and K-09 were done using conventional fading correction method (Huntley and
Lamothe, 2001). For samples showing over-dispersion (OD) ≤20%, central age model (CAM)
has been used for estimation of equivalent dose (De) (Bailey and Arnold, 2006) instead of
RMM-based De estimation as prescribed by Chauhan and Singhvi, (2011), useful for samples
having higher over dispersion (Table 2). For samples K-16 and K-17 having high OD value,
minimum age model (MAM) has been used. Details of sample preparation are provided in
supplement.
The dose rate was estimated using online software DRAC (Durcan et al., 2015) from the data of
Uranium (U), Thorium (Th) and Potassium (K) measured using ICP-MS and XRF (Table 1) in
IISER Kolkata. The estimation of moisture content was done by using the fractional difference
of saturated vs. unsaturated sample weight (Table 1).
**4. Results**

*4.1.Field observations and measurements*
The Chenab River has deeply incised the KW (Fig. 3b and 3e). The LHS rock units
exposed within the KW are mainly composed of fine-grain Quartzites and phyllites with
occasional schists in between. (Steck, 2003; Gavillot et al., 2018). The Lesser Himalaya has been
suggested to be an asymmetric antiformal stack with a steeper western flank (dip: 70°/west)
(Fig.2c). The KW is surrounded by rock units related to the Higher Himalayan high-grade
metasedimentary sequence, mainly garnet-bearing mica schists and gneisses. Higher Himalayan
rocks close to the western edge of the KW form a klippe with a southwest-verging MCT at its'
base. The KT, southern structural boundary of the window margin accommodating the
differential exhumation between window internal and surroundings, is expressed as highly
deformed sub-vertical shear bands.
Along the traverse of the Chenab River through the KW and further downstream, two
prominent stretches along the Chenab River ~20 and ~25-30 km length are characterized by
steep channel gradient associated with a large number of rapids (Fig.3b). These steep segments
are also characterized by a very narrow channel width (< 30m) (Fig.3b, 3e). The steepened
segments define knickzone rather than a single knickpoint. The knickzones K1 in the trunk
stream as well as in the tributaries are hosted over bedrock gorges. Although the knickzone K2
pass through a series of old landslides (around Kishtwar town), the rapids have all formed in
bedrock channel. Therefore, neither K1 nor K2 appears to be related to damming by recent
landslides or other mass movements. The eastern margin of the KW is characterized by a wide
'U-shaped' valley filled with thick sand layers and coarser fluvioglacial sediments (Fig. 3a)
where the Chenab River incises through this Late Pleistocene fill at present.
The rock strength data taken along the Chenab trunk stream portray large variations (R-
value ranging from 28 to 62) across different morphotectonic segments (Fig.7e). Within the KW,
Lesser Himalayan phyllites and schists have low R values (30-35); however, the low-strength
schists and phyllites are sparsely present and therefore, they are ignored while plotting the
regional rock strength values in Fig.7e. The dominant Lesser Himalayan quartzites in KW, as
well as the granitic intrusives in the eastern part of the KW, shows very high R values of 55-62
and 51-56 respectively (*Fig. 7e*). Compared to the high R values in the KW, the Higher
Himalayan metasediments show low strength (R: 35-45) till the point L2 (*Fig. 3b*). However,
near the western margin of the KW, the migmatites of Higher Himalayan domain show high rock
strength (R value: 58±3) (Fig.7e). The rock strength increases within the Haimanta Formation
(R: 44±2) further downstream until it reaches the MCT shear zone at the southern boundary of
the Main Himalayan orogen. The R-value in the frontal Lesser Himalaya is moderate (R: 41±2).
The Higher Himalayan sequence dips steeply away from the duplex (~65° towards west)
*(*Fig.2a, 8a*).* The frontal nappes of the Lesser Himalaya expose internally-folded greenschist
facies rocks. Although at the western margin of the duplex, the quartzites stand sub-vertically,
the general dip amount reduces as we move from west to east for the next ~10-15 km (*Fig. 8*).
Near the core of the KW, we observed deformed quartz veins of at least two generations, as well
as macroscopic white mica. Near the core of the window, where the river is also very steep and
narrow, the rock units are also steeply-dipping towards the east (~60-65°) and are extremely
nearly isoclinal and vigorously deformed at places (Fig.2d, 2e). Towards the eastern edge of the
window, however, the quartzites dip much gently towards the east (~25-30°) and much lesser
folding and faulting have been recognized in the field.

The E-W traverse of the Chenab River is completely devoid of any sediment storage.

However, along the N-S traverse parallel to the western margin of the KW, ~150-170m thick
sedimentary deposits are transiently-stored over the steeply-dipping Higher Himalayan bedrock.
The first study on sediment aggradation in Middle Chenab valley (transect from Kishtwar to
Doda town) was published by Norin (1926). He argued the sediment aggradation in and around
the Kishtwar town is largely contributed by fluvioglacial sediments and the U-shaped valley
morphology is a marker of past glacial occupancy. We partially agree to the findings of Norin
(1926) and Ul Haq et al., (2019) as we observe >100m thick fluvioglacial sediment cover
unconformably overlying the Higher Himalayan bedrock along the N-S traverse of the Chenab
River. The fluvioglacial sediments included alternate layers of pebble conglomerate and coarse-
medium sand. The pebbles are moderately rounded and polished suggesting significant fluvial
transport. Our field observations suggest that the fluvioglacial sediments have been succeeded by
a significant volume of hillslope debris. The thickness of the debris-flow deposits is variable.
The hillslope debris units contain mostly coarse-grained, highly-angular, poorly-sorted quartzite
clasts from the frontal horses of the Lesser Himalayan Duplex. The town of Kishtwar is situated
on this debris flow deposit (Fig.9). Along the N-S traverse of the Chenab, we have observed at
least two epigenetic gorges lying along the main channel (Fig. 3d). The active channel has
incised the Higher Himalayan bedrock and formed strath surfaces. We have identified at least
three strath surface levels above the present-day river channel, viz., T1 (280±5 m), T2 (170-175
m) and T3 (~120±5 m), respectively (Fig.3g, 10a).

**4.2. Results from morphometric analysis**

*4.2.1.   Steep stream segments and associated knickpoints*

The longitudinal stream profile along the Chenab River does not portray a typical

adjusted concave-up profile across the Himalaya (Fig. 6). We observe breaks in slope and
concavity at least at six localities within a ~150 km traverse upstream from the MBT across the
KW. These breaks are defined as knickpoints or knickzones depending on their type
characteristics. The slope breaks define the upstream reaches of the steep stream segments. The
basinwide steepness indices span from ~30- >750 $m^{0.9}$ across the study area (Fig. 5d). We
assigned a threshold value of $k_{sn}$>550 for the steepest watersheds/ stream segments. Along the
traverse, the major knickpoints are L1 (~1770m), K1 (~1700m), K2 (~1150m) and L2 (~800m)
respectively (Fig.6).

Already Nennewitz et al., (2018) had proposed a high basin-averaged $k_{sn}$ value of > 300

in the KW. Here in this study, we worked with a much-detailed DEM and stream-specific $k_{sn}$
allocation (Fig.7d), as well as a basinwide steepness calculation. Our results corroborate with the
earlier findings, but, predict the zone of interest in greater detail. It is important to note that by
setting a higher tolerance level in the 'knickpointfinder' tool in Topotoolbox, we have managed
to remove the DEM artifacts from consideration (Schwanghart and Scherler, 2014).

*4.2.2.   Channel width and valley morphology*

The channel width of the Chenab River is on average low (30-60m) within the core of the

KW (Fig. 3b, 7b), and the low channel width continues till the Chenab River flows N-S along the
western margin of the KW. However, there are a few exceptions; upstream from the knickpoint
L1 in the Padder valley (in which the town of Padder is located), the channel widens (width ~80-
100m) and the channel gradient is low (*Fig. 3a*). The second instance of a wider channel is seen
upstream from knickpoint K2, where there is a reservoir for the Dul-Hasti dam. Downstream
from K2 within the Higher Himalaya, the channel width ranges from 50-70 m. However, towards
the lower stretches of the N-S traverse, the width is even lower (16-52m). The river width
increases to 100-200m as Chenab River takes a westward path thereafter. The river width
increases beyond 300m until it leaves the crystalline rocks in the hanging wall of the MCT and
enters the Lesser Himalaya in the hanging wall of the MBT across the Baglihar dam. Within the
frontal LH, the channel width is again lowered (50-80 m).
*4.2.3.  Changes in specific stream power (SSP)*
Discharge-normalized SSP data calculated from the upstream stretches and the
knickzones, K1 and K2 show major increase in SSP within the steep knickzones. The increase in
SSP from upstream to the knickzones K1 and K2 are 4.44 and 5.02 times, respectively (Table 1).
Such high increase in SSP is aided by steepening of channel gradient (Fig.7c) and narrowing of
channel bed (Fig.7b).
**4.3. Luminescence chronology**
The results for the luminescence chronology experiment are listed in Table 2. Samples
collected from the fluvioglacial sediments overlain by debris flow deposit, namely as, K07, K08
and K09 yield IRSL ages of 104.5±5.9 kyr, 114.4±6.3 ky, and 119.2±6.8 kyr, respectively.
Fading corrections done for samples K07 and K09 yield the correction factors (g%) of 0.89 and
1.11 respectively. The sample K08 has not been treated for fading correction, but for easier
understanding, we have assumed a constant sedimentation rate between the samples K07 and
K09 and extrapolated the 'fading-corrected' age for K08. The oldest sample K09 (132±7 kyr)
(fading-corrected IRSL age) is succeeded by samples K08 (126±6 kyr) and K07 (113±6 kyr)
respectively. The finer fraction of the hillslope debris overlying the fluvio-glacial deposits yield
OSL ages of 81.1±4.6 kyr (K02) and 85±5 kyr (K11) (Fig.6). OSL samples taken from sparsely-
preserved sediment layers above the T3 strath surface shows heterogeneous bleaching and hence
we provide a minimum age of 22.8±2.1 kyr (sample K16) and 20.5±1.0 kyr (sample K17). One
sample taken above T1 strath level is saturated and shows a minimum age of 52.1±2.8 kyr
(sample K18) (Table 2). OSL samples K01 and K06 taken from sand layers sitting atop the
Higher Himalayan bedrock straths near the town of Doda portray depositional ages of 49.8±2.9
kyr and 51.6±2.4 kyr, respectively (Table 2).

**5. Discussions**

Morphometric parameters are widely used as indicators of active tectonics and transient

topography (Kirby and Whipple, 2012; Seeber and Gornitz, 1983). Many studies have used
morphometry as a proxy for understanding the spatial distribution of active deformation across
certain segments of the Himalayan front (Malik and Mohanty, 2007; van der Beek et al., 2016;
Nennewitz et al., 2018; Kaushal et al., 2017). More importantly, some studies have integrated
morphometric analysis with chronological constraints to assess the spatial and temporal
variability in deformation within the Sub-Himalaya (Lave and Avouac, 2000; Thakur et al.,
2014; Vassalo et al., 2015; Dey et al., 2016; Srivastava et al., 2018). All these studies have
shown that morphometric indicators can also be used for a qualitative estimate of changes in
uplift rate or spatial variations of deformation, even in the Sub-Himalayan domain where the
rivers are often alluviated due to high sediment load (Malik and Mohanty, 2007). Therefore,
using morphometric indices to examine some prospect areas and using their relative difference as
a proxy of relative changes in faulting and differential uplift as well as connecting these regions
with nearby regions having chronological constraints on short-intermediate timescale
deformation, is a potent option, when applied carefully.
The KW exhibits younger Apatite fission-track cooling ages (~ 2-3 Myr) as compared to
the surrounding Higher Himalaya, which have been interpreted as the result of rapid exhumation
of the LH duplex over $10^6$-year timescale (Gavillot et al., 2018). However, we lack any
measurements of deformation across the KW over the $10^3$-$10^5$-year timescale. With the existing
AFT data and assuming that no major changes of the deformation regime have taken place since
the Quaternary, we may well use it for calibration of morphometric proxies and interpolate these
estimates to regions, where no thermochronological constraint exists. Thus, we have come up
with a morphometric analysis of the terrain and combined those results with existing chronology
and structural data as a proxy for the spatial distribution of faulting and fault patterns.

**5.1. Knickpoints and their genesis**
Already Seeber and Gornitz (1983) showed that the Chenab River is characterized by a
zone of steep channel gradient in the vicinity of the KW. Thiede and Ehlers (2013) demonstrated
a strong correlation between steeped longitudinal river profiles and young thermochronological
cooling ages, suggesting recent focused rock uplift and rapid exhumation along many major
rivers draining the southern Himalayan front. Although, it is still an open debate whether uplift
and growth of the LH Duplex are triggered solely by slip over the crustal ramp of the MHT or
additional out-of-sequence surface-breaking faults are augmenting it (Herman et al., 2010; Elliot
et al., 2016; Whipple et al., 2016).

The longitudinal profile of the lower Chenab traverse (below ~2000 m above MSL) is

punctuated by two prominent stretches of knickpoint zones (Fig.6). Below we will discuss the

potential cause of formation of those major knickpoints in the context of detailed field

observation, of existing field-collected structural and lithological data, geomorphic features, rock

strength and channel width information (Fig.7).

*5.1.1.  Lithologically-controlled knickpoints*

The Himalayan traverse of the Chenab River is characterized by large variations in

substrate lithology and rock strength (Fig.1, Fig.7e). These variations have inflicted their 'marks'

on the river profile. An instance of soft-to-hard substrate transition happens across the knickpoint

L1, lying downstream from the Padder valley, at the eastern edge of the KW (Fig.2a). Across L1,

the river enters the LH bedrock gorge (R value> 50) after exiting the Padder valley filled with

unconsolidated fluvioglacial sediments (Fig. 3a). A similar soft-to-hard substrate transition is

observed upstream from the MCT shear zone. The corresponding knickpoint L2 represents a

change in lithological formation from the sheared and deformed Higher Himalayan crystalline (R

value~35-40) to deep-seated Haimantas (R value~40-50). There is no field evidence, such as

fault splays or ramps, in support of L2 to be a structurally-controlled one.

*5.1.2.  Tectonically-controlled knickpoints*

Compiling previously-published data on regional tectonogeomorphic attributes (Gavillot

et al., 2018) with detailed field documentation of structural styles and tectonic features; we

identified several proxies to constrain spatial variability in rock uplift and faulting across the

KW. We have found at least two instances where knickpoints are not related to change in

substrate, nor are they artificially altered.

The knickzone K1 (~1700 m above MSL) represents the upstream reach of a steepened
stream segment of run-length ~18-20 km. The steep segment represents a drop of ~420m of the
Chenab River across a run-length of ~15-20 km (Fig.8c). The upstream and downstream side of
K1 is characterized by a change in the orientation (dip angle) of the foliation of the LH bedrock
(Fig. 2f, 2g, 8). Across K1, the dips of the foliation planes change from ~30° to ~60-65° towards
east. K1 also reflects a change in the channel width (Fig. 7b). The steep segment exhibits a
narrower channel through the core of the KW. Near the end of the steep segment, we observed
intensely-deformed (folded and fractured) LH rocks (Fig.2d, 2e). There can be two main
possibilities for such observation – (1) it may be an active out-of-sequence fault or (2) it may be
an inactive fault that defines the floor-thrust of any of the numerous duplex nappes. We do not
find any conclusive evidence of recent activity along this deformed zone, which passively
favours the second possibility. On the contrary, the observed changes in the geomorphic indices
along with stretch of the knickzone K1 and observed increase in the bedrock dip angle may well
be explained by a ramp on the basal decollement. This explanation is supported by the existence
of mid-crustal ramps in the balanced cross-section from Gavillot et al., (2018). However, the
structural orientation of the rocks (Fig.8a) differ considerably than the proposed LH duplex in
Gavillot et al., (2008) raising questions about the duplex-model. Our field observations are
supported by works from Fuchs (1975), Frank et al., (1995) and Stephenson et al., (2000) who
argued against duplexing of multiple thrust nappes and favoured internal folding of Chail nappe
for the growth of the KW. Therefore, we cannot clearly comment whether K1 represents a
transition from flat to ramp of the MHT or is it indeed an active out-of-sequence thrust-ramp.
On the other hand, the other knickpoint K2 nearly coincides with the exposure of the KT
(Fig.6). K2 cannot be a lithologically-controlled knickpoint as it reflects a hard-to-soft substrate
transition from LH rocks (R value> 50) to HH rocks (R value< 45). We acknowledge that just
across the point K2, there are some strong leucosomatic layers within the migmatites (R: 58±3),
but in general, the migmatites are also deformed. The rock strength measurement was not done
in the multiply-fractured units as it would show inaccurate values. In the longitudinal profile, K2
does not represent a sharp slope break because the downstream segment runs parallel for ~25-30
km and not perpendicular to the orientation of all major structures of the orogen, including the
KT. Therefore, we performed an orthogonal projection of the E-W trending traverses of the
Chenab River and estimated an orogen-perpendicular drop of the Chenab across K2 (*Fig. 8c*).
The truncated profile across K2 shows a drop of ~230m of the channel across an orogen-
perpendicular run-length of ~5 km. The orogen-parallel stretch of the river exhibits narrow
channel width (<30-35m) through moderately hard HH bedrock (R-value: 35-45). The tributaries
within this stretch form significant knickpoint at the confluence with the trunk stream (Fig.3f).
These evidences hint towards a rapid uplift of the HH rocks near the western margin of the KT
and are possibly related to the presence of another crustal ramp emerging from the MHT
(*Fig.8d*). Although we didn't find any field evidence of regionally-extensive fault along the N-S
traverse of the Chenab River, similar topographic and morphometric pattern can be caused by an
active out-of-sequence fault.

Both the knickzones, K1 and K2 portray transiently-high specific stream power values

(Table 1). This signifies the fact that the knickzones are undergoing much rapid fluvial incision
than the rest of the study area. If we consider the fluvial incision as a proxy of relative uplift
(assuming a steady-state), we infer that the knickzones define the spatial extent of the areas
undergoing differential uplift caused by movement on the fault ramps.
*5.1.3.   Knickpoint marking epigenetic gorge*

Epigenetic gorges are common geomorphic features in the high-mountain landscape

(Ouimet et al., 2008). Epigenetic gorges form when channels of a drainage system are buried by
sediment aggradation and during subsequent re-incision, a new river channel is incised. The N-S
traverse of the Chenab River is largely affected by hillslope sediment flux (paleo-landslides and
debris flow) from the steep eastern flank. The knickpoint K3 situated near the village of Janwas,
mark one such instance of epigenetic gorge where the paleo-valley has been filled initially by
fluvioglacial sediments and the channel abandonment was caused by landslides and hillslope
debris flow prior to80 kyr (Fig.4b, 4c).
**5.2. Sediment aggradation in Chenab valley**
The Chenab valley records a net sediment aggradation since the onset of the last glacial-
interglacial cycle till ~80 kyr. Fluvioglacial outwash sediments range from ~110-130 kyr,
whereas the hillslope debris ranges from ~90 to ~80 kyr (Table 2). The chronology of the
sediments is in agreement with the overall stratigraphic order of the sediments. We observe net
fluvial incision and formation of bedrock strath surfaces since ~80 kyr (Fig.10).
**5.3. Drainage re-organization and strath terrace formation along Chenab River**
Hillslope debris flow from the high-relief frontal horses of the Lesser Himalayan Duplex
overlies the fluvio-glacial sediments stored beneath the Kishtwar surface. We argue that the
hillslope debris are paleo-landslide deposits which intervened and dammed the paleo-drainage of
the Chenab River, which might have been flowing through an easterly path than now (Fig.9).
The Maru River, coming from the northwestern corner of our study area was also joining the
Chenab River at a different location (Fig.9). Our argument is supported by field observation of
thick silt-clay layer in the proposed paleo-valley of the Maru River (Fig.9a, 9c). OSL sample
(K18) from the silt-clay layer is saturated and hence only provide the minimum age of 52±3 ky.
We suggest that the hillslope sediment flux dammed the flow of the Chenab River and also
propagated through the aforesaid wind-gap of the Maru River. The decline in the depositional
energy has resulted into reduction of grain-size. Post-hillslope debris flow, the Chenab River also
diverted to a new path. The new path of the Chenab River upstream from the confluence with the
Maru River is defined by a very narrow channel flowing through the Higher Himalayan bedrock
gorge (Fig.7b). Downstream from the confluence, we are able to identify at least three levels of
strath terraces lying at heights of ~280-290m (T1), ~170m (T2) and ~120m (T3), respectively
(Fig.3g,10a). Our field observation suggests that the formation of the straths is at least ~52 kyr-
old. The luminescence chronology samples in this study belong to the ~150-170m-thick soft
sediments that are stored stratigraphically-up from the T1 strath level. Our field observations and
chronological estimates suggest that the renewed path of the Chenab River, must have been
formed post the hillslope debris flow ~80-90 kyr but before 52 kyr.
**5. 4. Rapid bedrock incision along Chenab River**
Considering the rate of excavation of softer sediments to be at least an order of magnitude
higher than the rate of bedrock incision (Kothyari and Juyal, 2013; Sharma et al., 2016), we
calculated the minimum bedrock incision rate at the western margin of the KW, using the height
of the T1 strath (~280±5 m) and the average age of the sediments from the Hillslope debris flow
deposit. It yields a minimum bedrock incision rate of ~3.1-3.5 mm/yr over the last 80-90 kyr.
Considering the saturated OSL sample from the paleo-valley, we estimated the maximum
bedrock incision since 52 kyr to be 5.1-5.5 mm/yr. Similarly, using the minimum age estimate of
the T3 terrace abandonment, we deduce a maximum bedrock incision rate of ~5.7-6.1 mm/yr
since ~21 kyr. However, further downstream, away from the KW, the average bedrock incision
rate derived from dated strath surfaces (~36±2 m high from the Chenab River) near the town of
Doda is 0.7±0.1 mm/yr (sample K01 and K06). We don't have bedrock incision rates from the
core and the eastern margin of the KW, as the core is devoid of sediment storage and the eastern
margin is filled with fluvioglacial sediments and the river is incising the fill.

**5.5. Findings in context with the previously-published data**

AFT-cooling ages by Kumar et al., (1995) showcased young cooling ages from the core

of the KW and its western margin (AFT ages: ~2-3 Myr) compared to the surroundings (AFT
age: 6-12 Myr). The high exhumation rates proposed by Gavillot et al., (2018) are based on using
a geothermal gradient of 35-40˚C/km in Dodson's equation assuming a 1-D model (Dodson,
1973). Additional data and thermal modeling are needed across the KW to constrain the
exhumation rates from vertical transect. However, lateral similarities of the regional topography
and age patterns along the Sutlej area, Beas and Dhauladhar Range (Thiede et al., 2017; Thiede
et al., 2009; Stübner et al., 2018) have yielded  similar exhumation rates in the range of 2-3
mm/y. Long-term exhumation rates from the NW Himalaya agree well with findings of
Nennewitz et al. (2018) who correlated the young thermochron ages with high basinwide $k_{sn}$
values suggesting high uplift rates over intermediate to longer timescales. However, a study from
the Sikkim Himalaya by Abrahami et al., (2016) portrays decoupling between long-term
exhumation rates and millennial-scale basinwide denudation rates. That study highlighted that in
high-elevation glaciated catchments the exhumation rates are significantly lower than millennial-
scale denudation rates. However, in case of the NW Himalaya, the proposed range of long-term
exhumation rates of ~3 mm/yr mm/y determined by Gavillot et al., (2018) agree with the
regional data pattern. Although the geomorphic implications on landscape evolution provide
resolution at shorter timescales than the low-T thermochron studies, our field observations and
analysis support a protracted uplift of the KW. Unless there has been an ongoing uplift, the
geomorphic signatures would have been subdued. Young low-T AFT ages (Kumar et al., 1995)
had been sampled from the steepened stream reaches, where the SSP is high (Table 1).
Interestingly, exhumation rates steepened stretches is nearly one order of magnitude higher than
that of the Higher Himalayan units in the klippe. Our estimates of SSP also reflect an increase by
~five times within the steepened stretches.
Deeply-incised channel morphology, steep channel gradients marked by knickpoints at
the upstream reaches in and around the KW could be explained by the presence of at least two
orogen-parallel mid-crustal ramps on the MHT (Fig.8d). Existence of two mid-crustal ramps has
already been shown through sequential balanced cross-sections for the last 10 Myr across the
Kashmir Himalaya (Gavillot et al., 2018). Translation on the MHT can impart differential uplift
of the LH duplex across the two mid-crustal ramps as ramps would show higher uplift/
exhumation. Here we provide more detailed information on structural styles across the KW
(Fig.8a, 8d). Our field observation questions the existence of multiple nappes forming a duplex
(Gavillot et al., 2018) and rather favors anticlinal doming of the pervasively-deformed Chail
nappe, as suggested by Fuchs (1975). We observe pronounced deformation at the core of the KW
(Fig. 2d, 2e) suggesting that this is related to active faulting, crustal buckling or internal folding
which maintain continuous rock-uplift forcing the Chenab River to incise and prevail the
steepened stretch of K1. Gavillot et al., (2018) proposed that translation on a mid-crustal ramp of
the MHT and no surface-faulting is driving the uplift at the core of the KW (Fig.8d). One
alternative explanation is the existence of a crustal fault-ramp emerging from the MHT that
triggers rapid exhumation of the hanging wall. In this case, out-of-sequence faulting causes high
relief, steep channel gradients and higher basinwide steepness indices over the ramp (Fig.7).
Similar ramps have been proposed on the MBT beneath the Dhauladhar Range (Thiede et al.,
2017) and in the east of the NW Himalaya (Caldwell et al., 2013; Mahesh et al., 2015; Stübner et
al., 2018; Yadav et al., 2019). Similar mid-crustal ramp (MCR-2) has been proposed for the
western margin of the KW by Gavillot et al., (2018).We don't have any direct field evidence of
regional surface-breaking faults which could be related to K2 knickzone. However, a rapid
fluvial incision and transient increase in morphometric parameter values probably justify the
existence of either a mid-crustal ramp or an out-of-sequence surface-breaking fault.
Our findings from the Kishtwar region of the NW Himalaya establish the importance of
morphometric parameters in the assessment of intermediate timescales of $10^4$-$10^6$ years. We can
resolve regional variations in the tectonic uplift and related landscape evolution by analyzing the
topography with high-resolution DEM.
Models explaining the spatial distribution of the high uplift zone in the interiors of the
Himalaya favor the existence of a mid-crustal ramp, which has variable dimension, geometry,
and distance from the mountain front along-strike of the Himalayan orogeny (Robert et al.,
2009). Nennewitz et al., (2018) have proposed that the million-year-timescale shortening
achieved in the interior of the Himalaya near the Sutlej-Beas area in the eastern Himachal
Pradesh is caused by accentuated rock uplift over a ramp at a mid-crustal depth of ~ 8-25 km on
the MHT. In contrast, studies from the Dhauladhar Range in the north-western Himalaya hints
the presence of deep-seated crustal ramp on the MBT and yielded a shortening rate of 3±0.5
mm/yr across the MBT over the last 8 Myr and absence of mid-crustal ramp (Deeken et al.,2011;
Thiede et al., 2017). The work by Gavillot et al. (2018) favors the existence of at least two mid-
crustal ramps beneath the KW (Supplementary Fig.B2). Their suggestion is in agreement with
very young AFT cooling ages (1-3 Ma) (Kumar et al., 1995) in the window (Fig.1a).Our data
further supports the idea of mid-crustal ramps beneath the Higher Himalayan domain across the
Kashmir and NW Himalaya (Webb et al., 2011; Gavillot et al., 2018; Nennewitz et al., 2018) and
possibly explains why the seismic hypocenters are clustered in the vicinity of the proposed ramp
of MHT. The seismicity is linked to the ongoing deformation of the Lesser Himalayan anticlinal
stack or duplex. These studies altogether point out the along-strike variation in the location of the
rapidly-uplifting crustal ramp with respect to the southern Himalayan front. The crustal ramp in
the nearby Kangra recess is located beneath the Dhauladhar Range at the main Himalayan front,
whereas, in the Himalayan transects situated towards the east and west of Kangra recess, the
ramps are located ~100km inside from the MBT. Topographic relief and basinwide mean ksn
distribution (Fig.5) hint towards the existence of a lateral ramp in between the Kangra and the
Jammu-Kashmir Himalayan transects. However, at this moment, we have no conclusive data in
support of this claim.

Detailed structural mapping and morphometric analysis using high-resolution DEM

provide important constraints on the spatial extent of deformation. We are able to resolve the
high-relief Kishtwar Window and the surroundings into two major steep orogen-parallel belts/
zones (Fig. 5e, 8d) - one at the core of the KW could be an active high-angle fault-ramp
emerging from the MHT or a crustal ramp; and the other one observed along the western margin
of the KW could be another ramp on the MHT or a surface-breaking fault. We suggest that this
has two major implications. One, the structural architecture of the MHT is variable along-strike
of the entire Himalayan orogen. The MHT may have a single or multiple mid-crustal ramps at
places and may have none in some transects. Alternatively, there may active out-of-sequence
faulting in the interiors of the Main Himalayan orogen. Secondly, the Kishtwar Window is still
growing and therefore could be the potential source of future seismic activity.

Although we speculate an out-of-sequence fault model for the growth of the KW, there is

an important concern regarding this model. Long-term crustal shortening estimated from low-T
thermochron data (Gavillot et al., 2018) and GPS-derived decadal shortening estimates (Stevens
and Avouac, 2015) imply steady crustal shortening of ~$13\pm1$ mm/yr. Assessment of late
Pleistocene-Holocene crustal shortening across the Sub-Himalayan domain of the Kashmir
Himalaya (Gavillot, 2014; Vassallo et al., 2015) suggests that the total Himalayan shortening
since late Pleistocene may have been accommodated only within the Sub-Himalaya; therefore,
there is no need of additional out-of-sequence faulting in the KW. However, this is again an
assumption that the cumulative crustal shortening rate is steady across different timescales.

**6. Conclusions**

Our field observation and the characteristics of terrain morphology match well with the

spatial pattern of previously-published thermochronological data and indicate that the Kishtwar
Window is undergoing active and focused uplift and exhumation at present, during intermediate
timescales, and in geological past since at least the late Miocene. By compiling all the results and
published records, we favor the following conclusions:
1.     The Chenab River maintains an over-steepened bedrock channel and a low

channel width irrespective of lithological variations across the KW and beyond,

suggesting ongoing rapid fluvial incision related to active tectonic rock uplift.

2.     Our field observations, morphometric analysis, and rock strength measurements

document that at least two of these major knickzones with steep longitudinal

gradients on the trunk stream are non-lithologic and are likely related to

differential rock uplift. The incision potential (specific stream power) in the

steepened stretches ~4-5 times higher than the surroundings.

3.      The differential uplift can be explained either by slip on the multiple ramps on the

MHT and exhumation of the duplex floor-thrust or by a combination of slip on the

MHT ramp and active out-of-sequence faulting. As of now, we do not have any

evidence for large-scale out-of-sequence faulting.

4.      Luminescence chronology of the transiently-stored sediments along the Chenab

River suggests that the valley had been overfilled by sediments of fluvio-glacial

origin as well as by hillslope sediment flux. Massive sediment aggradation during

~130-80 ky led to drainage re-organization and bedrock incision leaving behind

strath surfaces.

5.      The late Quaternary bedrock incision rates near the western margin of the KW are

high 3.1-3.6 mm/y while away from KW, the incision rates are low ($< 1$ mm/y).

We argue that the high fluvial incision rate can potentially be linked to

accommodation of crustal shortening either by growth of the duplex or by active

out-of-sequence faulting near KT.

To summarize, our new study reinforces the importance of detailed field observation, and
morphometric analysis in understanding the neotectonic framework of the interiors of the
Himalaya. With additional chronological evidence from the transiently-stored sediments, we
showcase high rates of bedrock incision in the interior of the western Himalaya, which could
potentially be indicative of tectonic control on landscape evolution. However, to solve the debate
of ongoing duplex-growth vs. active out-of-sequence faulting, we would require more field data
on active structures and chronological constraints on deformation rates across potentially-active
structures.
**Appendix**
Additional maps, figures on morphometric analysis and luminescence dating are listed in
Appendix A. Data of rock strength measurements provided in Table C1. Luminescence sample
processing is elaborated in Appendix B.
**Code availability**
Authors used open-source codes of Topotoolbox and Topographic Analysis Kit Toolbox
for this study.
**Data availability**
Field data are already provided in Appendix 1. Additional data on luminescence dating
can be provided on request.
**Sample availability**
Samples used for luminescence dating are already mostly-destroyed, therefore it is
beyond sharing.
**Author contribution**
S.Dey, the first author , this work and completed the fieldwork, sample processing,
measurements and writing of this manuscript. R. Thiede helped in fieldwork, discussion and
writing of this manuscript. A. Biswas performed the initial morphometric analysis. N.Chauhan
helped in measurement of luminescence signal and assessment of the data. P.Chakravarti
performed the channel width calculations and compiled the rock strength measurements. V. Jain
helped in discussion and writing of the manuscript.
**Competing interests**
The authors declare that they have no conflict of interest.

**Acknowledgments**

This study is funded by the DST INSPIRE faculty fellowship program by the Department
of Science and Technology, India (grant #DST/INSPIRE/04/2017/003278), and IIT Gandhinagar
post-doctoral research fund (IP/IITGN/ES/SD/201718-01). Thiede is supported by German
Science Foundation (grant # DFG TH 1317-8 and 9). We thank M.K.Jaiswal and M.Rawat for
providing the elemental analysis. We thank Shambhu Das, Avi Das, Niklas Schaaf, Akashsingh
Rajput and Chamel Singh for their assistance during fieldwork. We also thank Soumyajit
Mukherjee, Rahul Kaushal and Shantamoy Guha for scientific inputs and comments on this
manuscript. We acknowledge A. Forte, Y. Gavillot and S. Hergarten for their constructive
reviews.

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

**Figure captions**

**Figure 1:** (a) An overview geological map of the western sector of the Indian Himalaya showing major lithology (modified after Steck, 2003 and Gavillot et al., 2018) and existing structures (Vassalo et al., 2015; Gavillot et al., 2018). The tectonic Kishtwar Window (KW) is surrounded by exposure of MCT, locally known as the Kishtwar Thrust (KT), and exposes the Lesser Himalayan nappes. The Lesser Himalaya forms a west-verging asymmetric anticline. Apatite fission-track (AFT) ages are adapted from Kumar et al., (1995). (b) A balanced cross-section of the NW Himalaya showing the general architecture of the Himalayan orogenic wedge (modified after Gavillot et al., 2018). Note that, beneath the KW, Gavillot et al., (2018) proposed the existence of at least two crustal ramps (MCR-1 and MCR-2) on the MHT, translation on which may have resulted in 3.2-3.6 mm/yr Quaternary exhumation rates across the KW.

**Figure 2:** Lithological units and structural orientations observed in the Chenab valley. (a) Steeply-dipping HHCS units near the western margin of the KW. (b) Highly-deformed migmatites at the base of the KT. (c) Sub-vertical quartzite slabs of Chail Formation exposed in the frontal horses of the LH Duplex (or, anticline). (d) Highly-deformed, sub-vertical and pervasively folded and compressed quartzite layers within the core of the KW, the base of stacked LH-nappes forming the hanging wall of the proposed surface-breaking fault (Fig. 8d). (e) A close-up view of the folded quartzite units. (f) Steeply-dipping units of granite which formed new penetrative foliation outcropping upstream from the fault-zone. (g) Further upstream from the fault-zone, the bedrocks are gentler in the eastern edge of the KW.

**Figure 3:** Figure 3: Geomorphic features observed along the Chenab River across the KW. (a) Where the Chenab River enters the KW, the major tributaries coming from the Zansar Range in the north are characterized by 'U-shaped' valley suggesting repeated glacial occupancy during

the Quaternary. The Chenab valley is unusually wide here providing space for transient storage
of glacial outwash sediments. The present-day River re-incises these sedimentary fills.
Photograph was taken near the town of Padder (cf. Fig.1a). (b) At the core of the KW, the
Chenab valley is V-shaped, steep The Chenab River is steep and maintains a narrow channel
width. (c) Highly-elevated fluvial strath surfaces are preserved in the vicinity of the town of
Kishtwar Fluvial incision observed along the N-S traverse of the Chenab River. Photograph was
taken from south of the Kishtwar town. The Kishtwar surface (~400m high from the river) is
underlain by ~150-170m thick sediment cover overlying the tilted Higher Himalayan bedrock.
The River has incised another ~240m bedrock in this section. (d) Epigenetic gorge formed along
the Chenab River in its' N-S traverse through the HHCS. The town of Drabshalla is built on the
hillslope deposits. (e) Chenab River maintained very narrow channel (width: ~20-25 m) through
moderately-strong HHCS rocks, suggesting tectonic imprint on topography. (f) Formation of
knickpoint at the confluence of the tributary with the trunk stream implying rapid fluvial incision
of the trunk stream. (g) Three levels of strath surfaces observed below the Kishtwar surface. The
strath levels are marked as T1 (~280m), T2 (~170m) and T3 (~120m). OSL dating of fluvial
sediments lying above the T3 surface yield a minimum depositional age of ~21.6±2.6 ky.
**Figure 4:** (a) Lithological distribution near the western margin of the KW (cf. Fig.8 for
location). Luminescence sample (OSL and IRSL) locations and respective depositional ages (in
kyr) are shown. Every sample except K16 and K17 are taken above strath level T1. K16 and
K17 are taken from above the T3 level. Note that, the ages reported in italics are minimum age
estimates. (b) A field photograph from the village Janwas, south of the town of Kishtwar,
showing the aggraded sediments lying above the Higher Himalayan tilted bedrock units. (c)
IRSL ages (in kyr) from the fluvioglacial sediments  and OSL age (in kyr) from the hillslope
debris units suggest the valley aggradation probably started at the transition of the glacial to
interglacial phase ~120-130 kyr and continued till ~80 kyr ago. (d) A close-up view (red
rectangle in fig.4c) of the tilted fluvioglacial sediment layers showing alternate conglomerate and
medium-coarse sand layers. (e) A ~3m thick fine sand layer within the hillslope debris yield
depositional age of ~86±5 kyr. Photograph was taken near the village Pochal, northwest of the
town of Kishtwar.
**Figure 5:** Regional variations in (a) topography, (b) topographic relief (moving window of ~4
km) (c) TRMM-derived rainfall (after Bookhagen and Burbank, 2006), and (d) Basinwide
Normalized steepness indices (ksn value) of the region shown dashed box in Figure 1a. (e)
Swath profiles (swath window: 50 km) along the line AB (cf. Fig.5a) demonstrate the orogen-
perpendicular variations in elevation, rainfall and ksn value. KW is characterized by high
elevation, high relief and high steepness, but low rainfall.
**Figure 6:** Longitudinal profile of the Chenab River show major changes in channel gradient
associated with knickpoints in the upstream. It illustrates the major changes in the channel
gradient extend over the full length of the KW and strongest changes are located in the core and
not at the margins of the window. We classified knickpoints on the basis of their genesis. The
substrate lithology along the River is shown.  Knickpoints caused by glacial occupancy (G1, G2
and G3) are adapted from Eugster et al., (2016), who reconstructed the timing of  maximum
glaciation and extent of glacial cover in source region of upper Chenab River basin during the
last glacial maximum. These knickpoints highlight the importance of glacial erosion in the high-
elevation sectors, especially in the northern tributaries of the Chenab River. Further in this study,
we focused on the area marked by red rectangle.

**Figure 7:** Along-river variations in (a) channel-elevation, (b) channel width, (c) channel gradient, (d) Normalized steepness index, and (e) rock-strength of non-fractured bedrock units (R-value taken by rebound hammer) till 165 km upstream from the MBT (point X, cf. Fig.1a). The mean R-value±σ for each rock type has been plotted against their spatial extent. We identified two distinct zones (K1 and K2) of high channel gradient and steepness index, which maintain low channel width despite the variable rock strength of the substrate. Knickpoint K3 may have been generated by the formation of the epigenetic gorge along the N-S traverse of the Chenab River (cf. Fig.3c). Knickpoints L1 and L2 mark the transition of a soft-to-hard bedrock substrate.

**Figure 8:** (a) Detailed structural data from the study area showing structural and lithological variations (modified after Steck, 2003; Gavillot et al., 2018). (b) and (c) orogen-perpendicular drop of the Chenab trunk stream across stretch 1 and stretch 2, respectively, showing transient increase in steepness over the K1 and K2 knickzone. The orthogonal profile projection method has been used in the case of K2 (cf. fig.7) to identify the width of the steep segment. (d) Comparison between two deformation models explaining the observed morphometric variations across the KW – (a) duplex-growth model (adapted from Gavillot et al., 2018) and (b) active out-of-sequence fault model.

**Figure 9:** A satellite image of the northern Kishtwar town showing the present-day flow-path of the Chenab River (cf. Fig.8 for location). Hillslope debris originated from the steep western margin of the KW (only made of massive white quartzites) and was deposited over fluvioglacial and glacio-lacustrine sediments and Higher Himalaya schists bedrock exposed below in the Kishtwar valley. Massive hillslope sediment flux impeded the paleo-drainage system leaving behind the paleo-valley of the tributary, the Maru River. Our interpretation of the paleo-drainage

is marked in a white dashed line.    (a) A view of the Kishtwar surface from the western margin
of the KW showing present-day gorge of the Chenab River and its tributary. The wind-gap
(paleo-valley) of the tributary is visible. (b) Thick clay-silt deposit in the wind-gap suggests
abandonment of river-flow. The OSL sample is saturated and hence only denotes the minimum
age of valley abandonment/ hillslope debris flow. (c) Overview picture of the frontal horses of
the LH duplex and the direction of debris flow towards the Kishtwar town. (d) Angular, poorly-
sorted clasts and boulders were observed at the base of the debris flow unit near the village of
Pochal, north of the Kishtwar town. The white quartzites of LH are exposed in the vicinity of the
Kishtwar Town (see satellite image) – only the eastern valley flank can have collapsed in the
past.
**Figure 10:** (a) A topographic and geomorphic profile across the Chenab valley drawn over the
Kishtwar Town. The valley aggradation by fluvioglacial and hillslope debris sediments was
succeeded by a fluvial incision which penetrated through the unconsolidated sediments of
thickness ~140-150m and incised Higher Himalayan bedrock by ~280±5 m, leaving behind at
least three recognizable strath surfaces with a thin late Pleistocene sediment cover. The three
strath surfaces are at 280±5 m (T1), ~170 m (T2), and ~120±5 m (T3) heights from the present-
day River. We assume that the present-day bedrock gorge has been carved since the deposition
of the glacio-lacustrine sediment deposits (~100-130 ky) and the hillslope debris (~90-80 ky)
onto former fluvial strath surface of Higher Himalayan Bedrock. The width of the fluvial strath
surface where the Kishtwar Town is located indicates that the river network had been dammed
earlier too. (b) Graphical representation of mean bedrock incision rates since 80 kyr. Age
constraints for T3 are shown in Fig. 4a. Based on relative heights and depositional ages of late
Pleistocene deposits, we propose a minimum and a maximum bedrock incision rate of 3.1-3.5
mm/y and 5.2-5.6 mm/yr, respectively. However, further downstream, the bedrock incision rates
calculated from bedrock straths farther downstream from the KW range 0.7-0.8 mm/yr.
**Table caption:**
**Table 1:** Calculations of change in specific stream power (SSP) values across the ramp and the
flat segments beneath the LH Duplex. We used a uniform discharge for SSP calculation.
**Table 2:** Sample locations, elemental concentrations, dose rates, equivalent doses and age
estimations for sand samples from Kishtwar valley.

Figures

1122                                    Figure 1

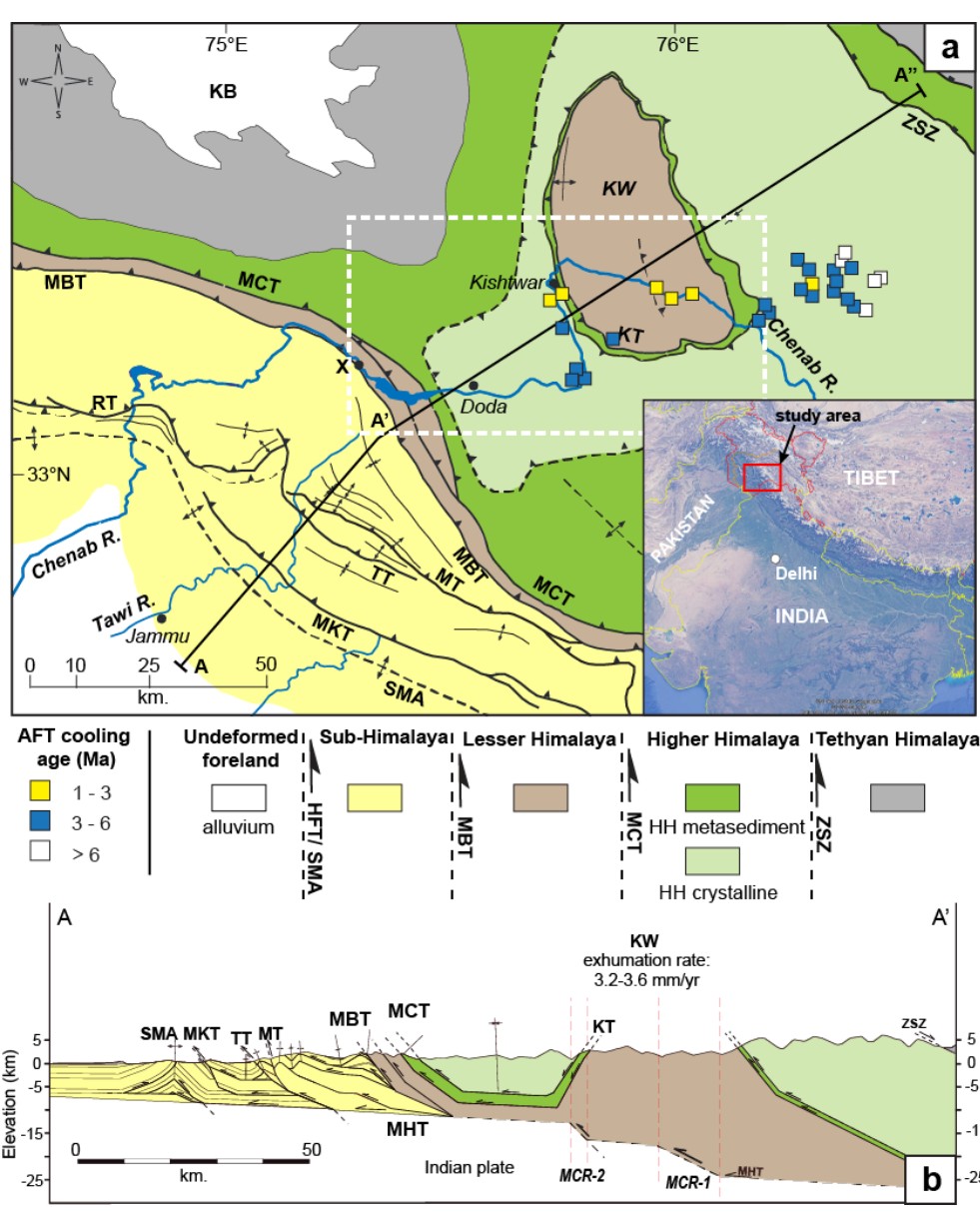




1126                                    Figure 2

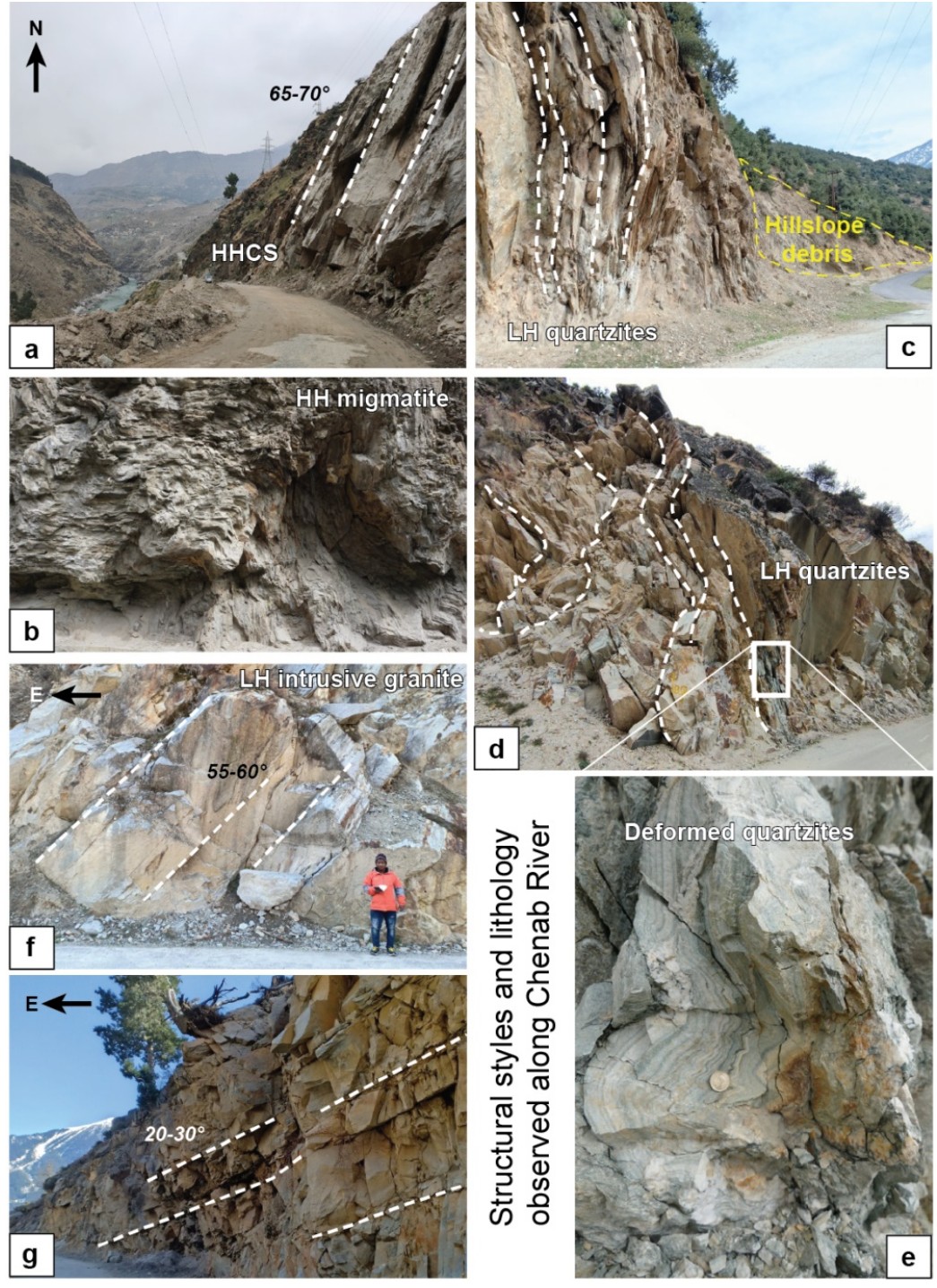



Figure 3

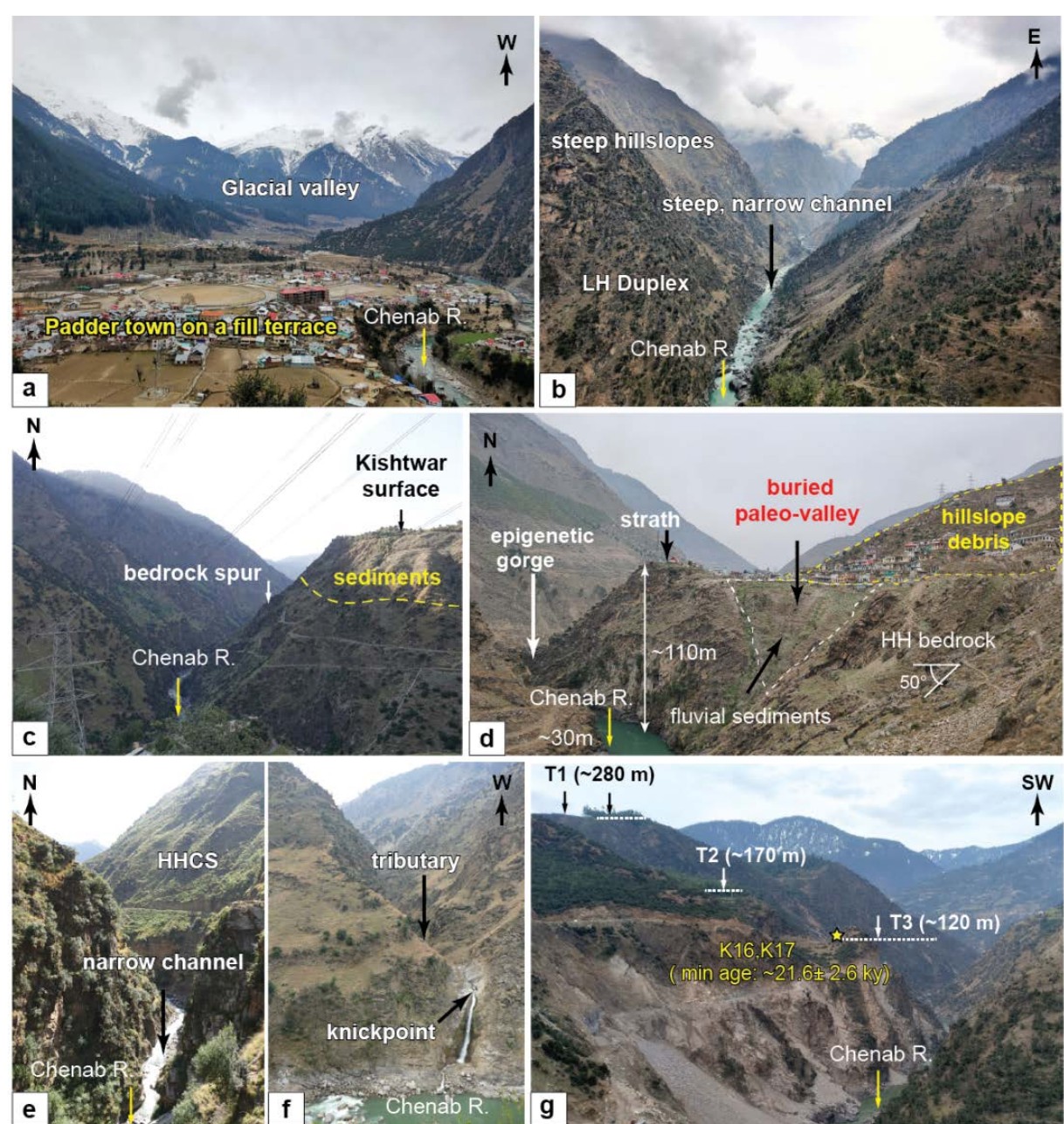


Figure 4

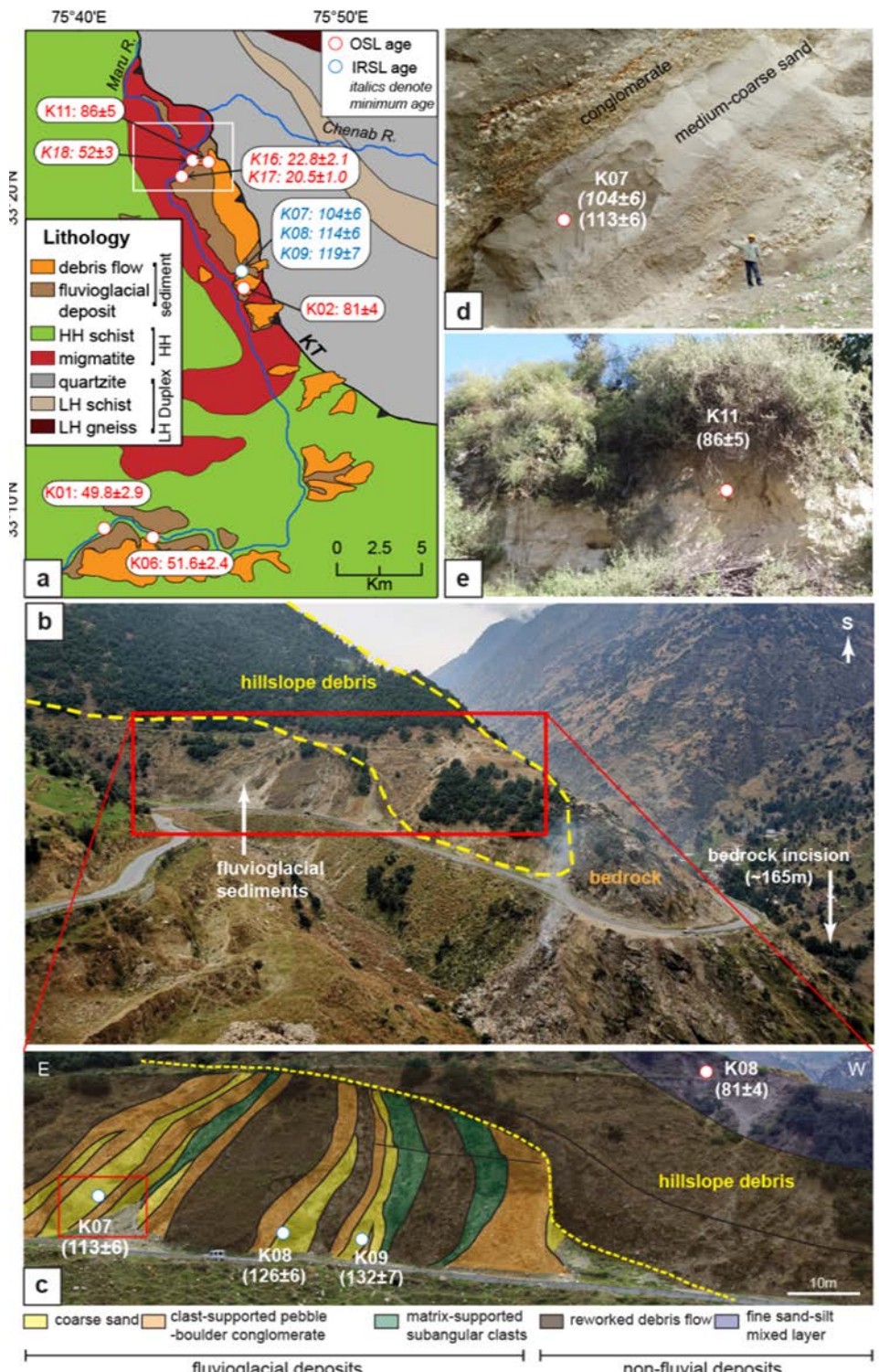



Figure 5

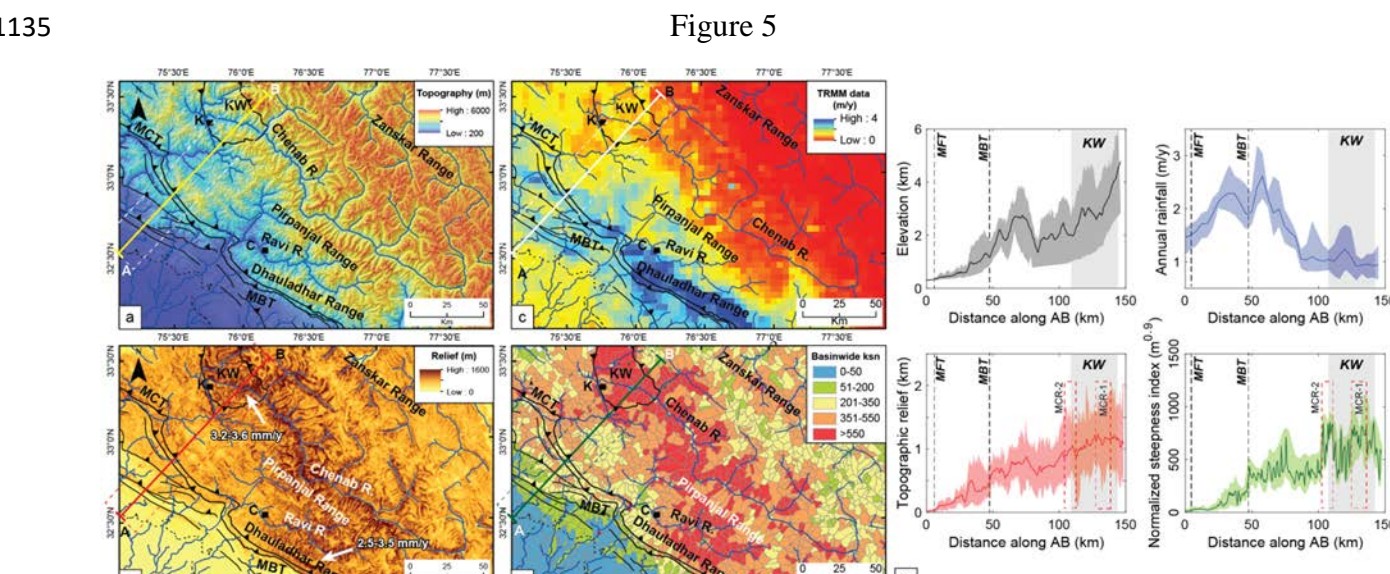

Figure 6

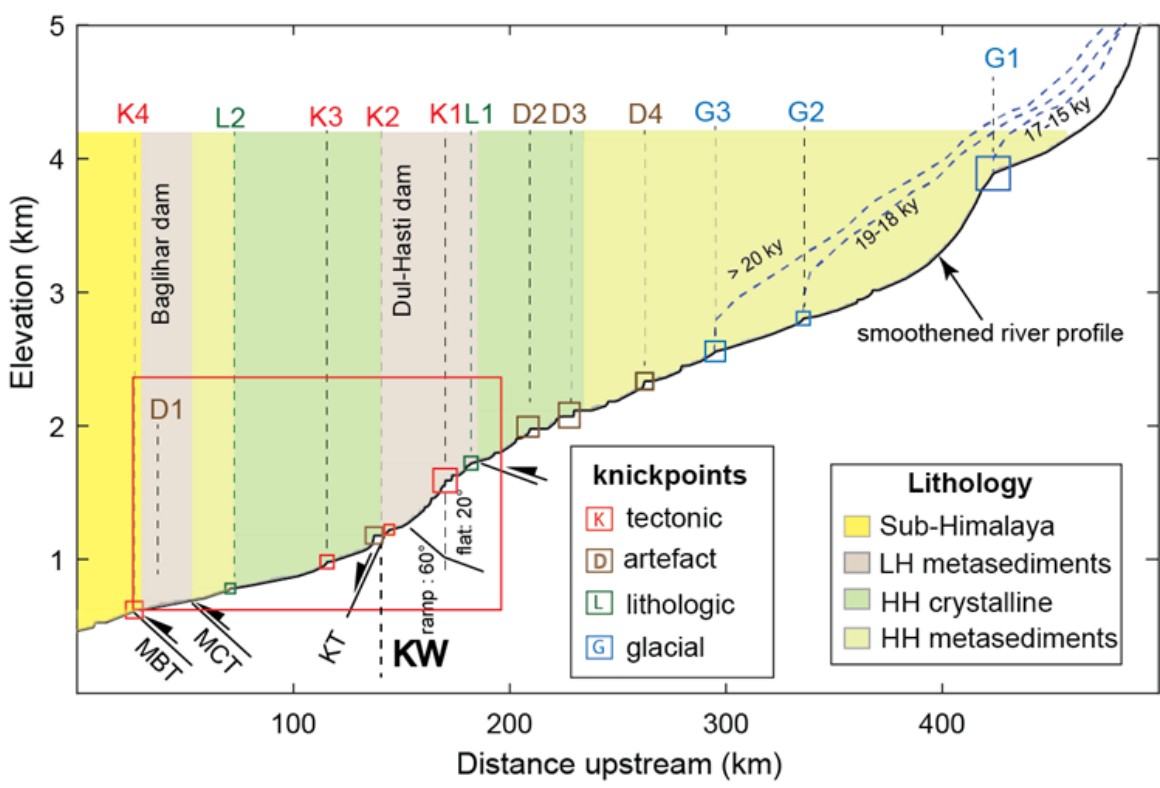

Figure 7

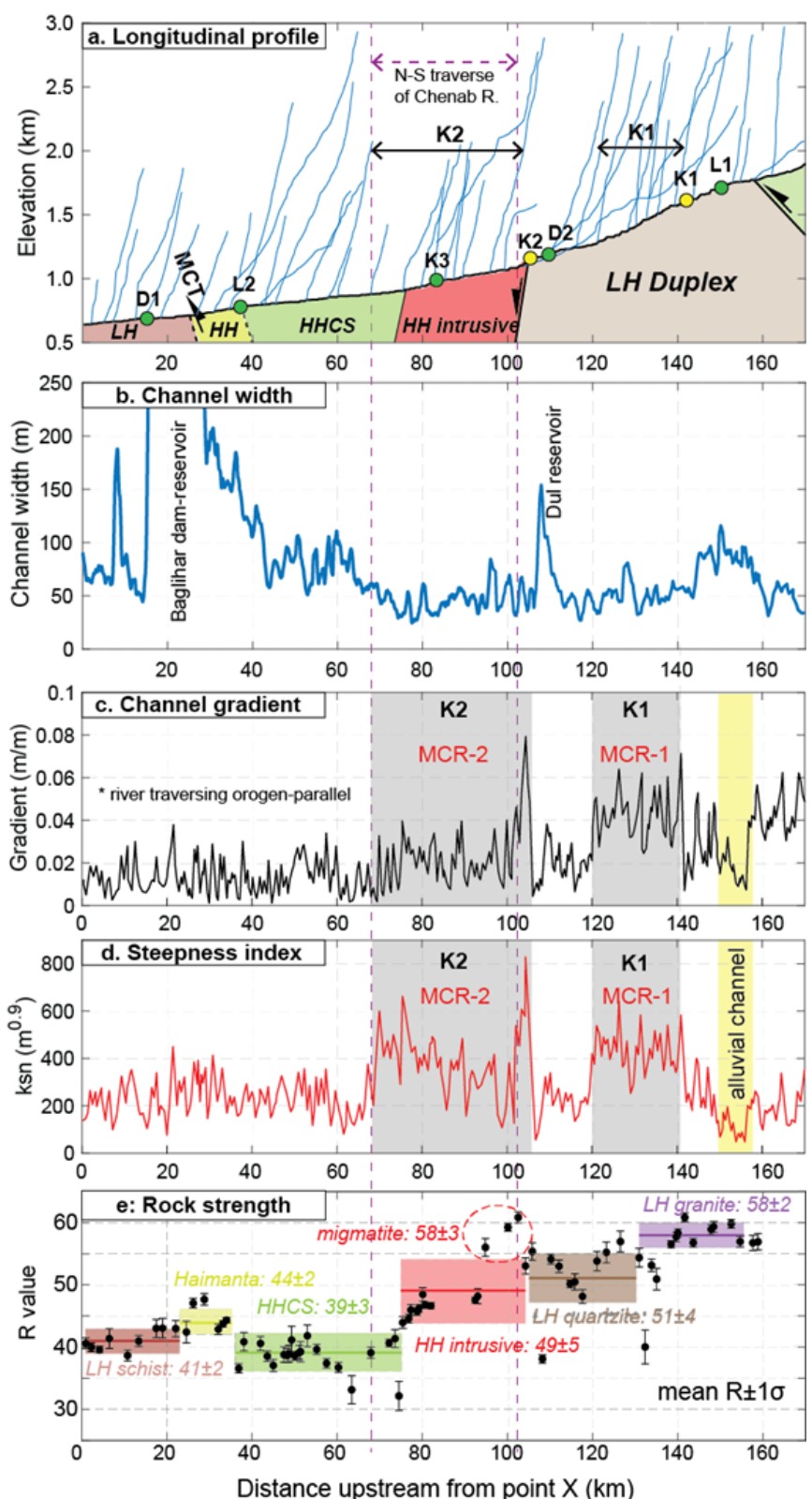

# Figure 8

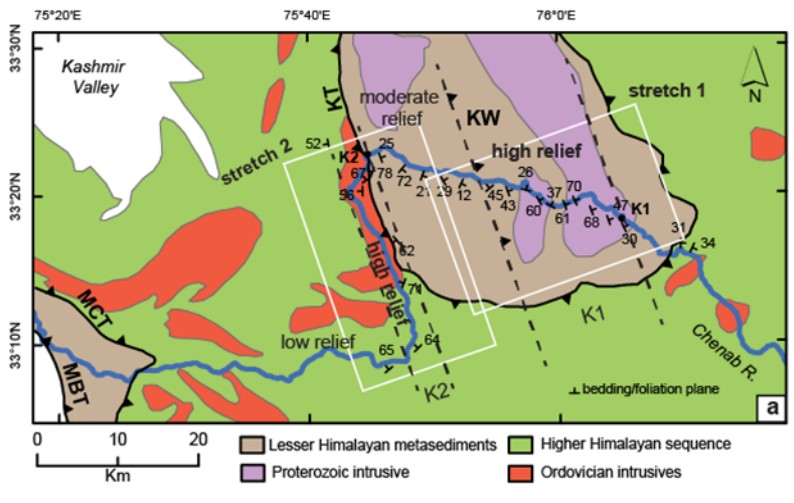

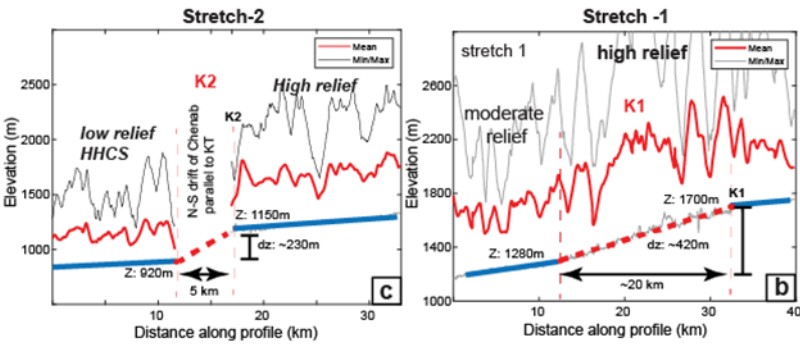

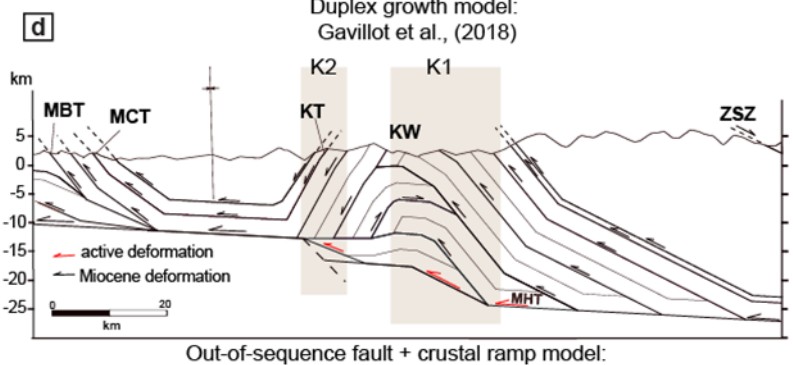

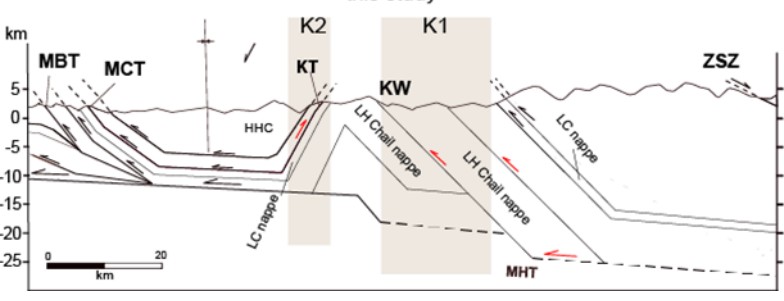

Figure 9

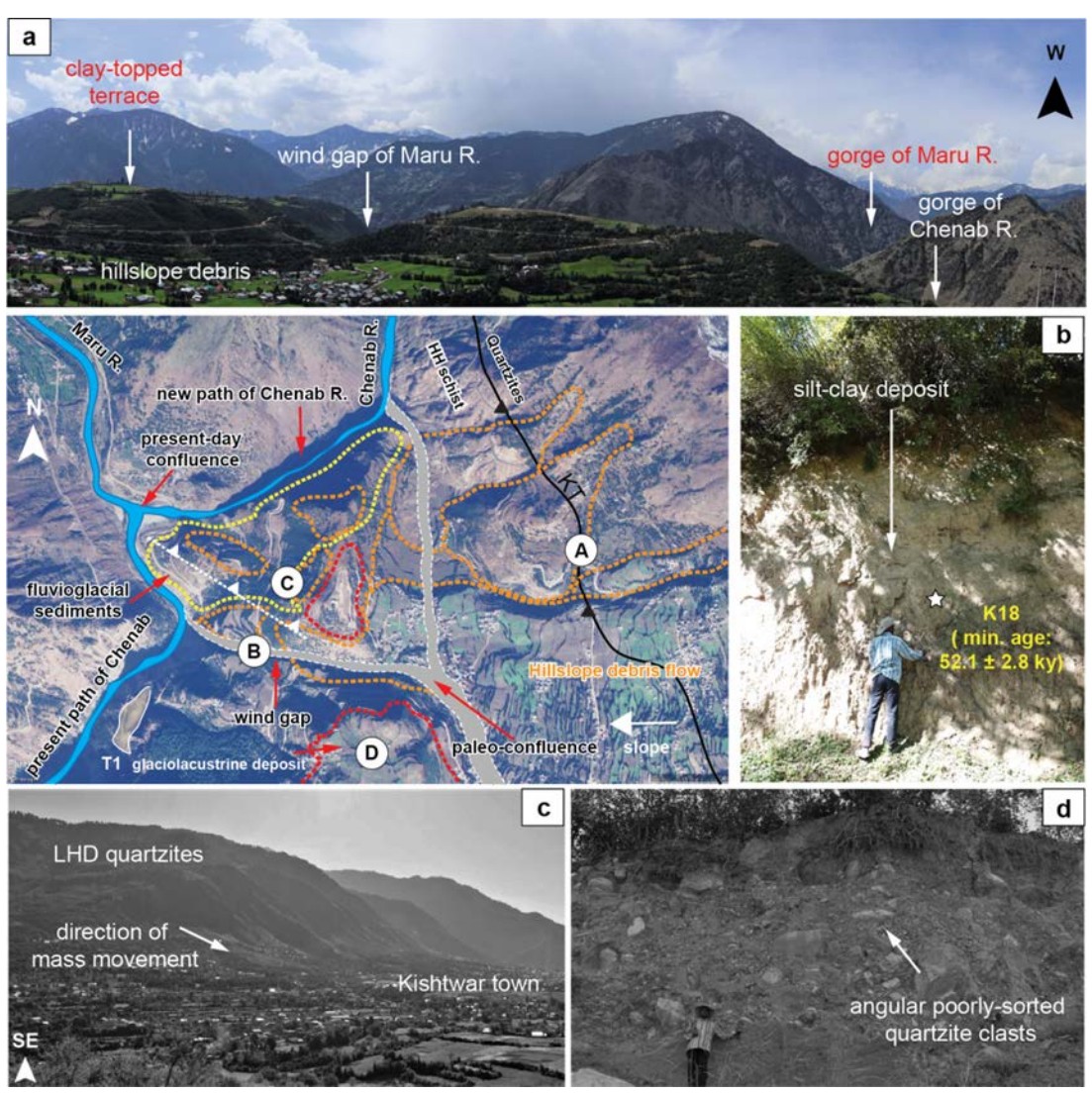

Figure 10

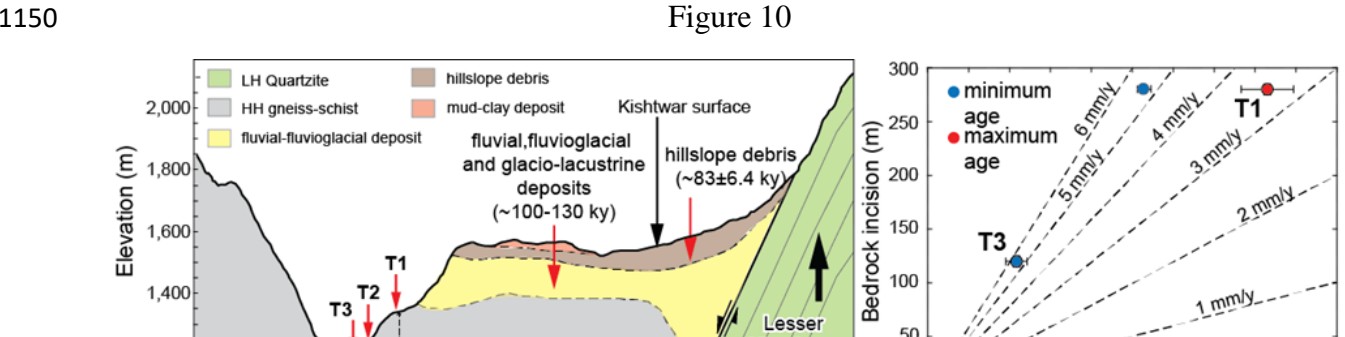

Table 1

| Parameter | flat 1 | ramp 1 | % change | ratio ramp 1:flat 1 | flat 2 | ramp 2 | % change | ratio ramp 2:flat 2 |
|---|---|---|---|---|---|---|---|---|
| average channel gradient (m/m) | 0.006 | 0.021 | 250.00 | 3.5 | 0.01 | 0.046 | 360 | 4.60 |
| average channel width (m) | 70 | 45 | -35.71 | 0.6 | 55 | 42 | -24 | 0.76 |
| *Specific stream power (SSP) | 0.000086 | 0.000467 | 444.44 | 5.4 | 0.000182 | 0.001095 | 502 | 6.02 |

* SSP calculated by assuming equal-discharge (Q)




Table 2

| Sample type | Sample name | Lat (°) | Long (°) | U (ppm) | Th (ppm) | K (%) | water (%) | Dose rate (Gy/ky) | De (Gy) | OD (%) | Age (ky) | fading correction | Corrected age (ky) |
|---|---|---|---|---|---|---|---|---|---|---|---|---|---|
| **using central age model** | | | | | | | | | | | | | |
| OSL | K02 | 33.29607 | 75.77619 | 3.8 | 7.2 | 0.46 | 6.1 | 1.74±0.02 | 141±8 | 19.5 | 81.1±4.6 | | |
| OSL | K11 | 33.35352 | 75.74649 | 3.1 | 12.7 | 2.41 | 6 | 3.97±0.09 | 341±19 | 16.8 | 85.7±5.1 | | |
| OSL | K01 | 33.15222 | 75.66323 | 2.9 | 13.2 | 2.03 | 9 | 3.88±0.04 | 193±11 | 22.1 | 49.8±2.9 | | |
| OSL | K06 | 33.15243 | 75.70609 | 3.4 | 18.0 | 2.17 | 5.4 | 3.97±0.05 | 205±10 | 14.4 | 51.6±2.4 | | |
| IRSL | K07 | 33.27780 | 75.76922 | 3.3 | 13.8 | 2.31 | 5.3 | 4.67±0.22 | 489±29 | 16.8 | 104.5±5.9 | 0.89 | 113±6 |
| IRSL | K08 | 33.27780 | 75.76922 | 3.5 | 16.9 | 1.97 | 5.6 | 4.61±0.23 | 528±38 | 20.5 | 114.4±6.3 | | |
| IRSL | K09 | 33.27780 | 75.76922 | 3.3 | 12.2 | 1.98 | 4.8 | 4.29±0.20 | 510±42 | 18.1 | 119.2±6.8 | 1.11 | 132±7 |
| **using minimum age model** | | | | | | | | | | | | | |
| OSL | K16 | 33.34873 | 75.73324 | 3.5 | 16.8 | 2.03 | 7.5 | 3.95±0.1 | 90±8 | 40 | 22.8±2.1 | | |
| OSL | K17 | 33.34873 | 75.73324 | 3.4 | 18 | 2.17 | 10.5 | 3.96±0.11 | 81±3.5 | 46 | 20.5±1.0 | | |
| **saturated sample** | | | | | | | | | | | | | |
| OSL | K18 | 33.35176 | 75.74325 | 3.3 | 18.7 | 2.61 | 4.5 | 4.36±0.13 | 227±14 | | 52.1±2.8 | | |


