# Peer review of "Implications of the ongoing rock uplift in NW Himalayan interiors"

_Earth Surface Dynamics, 2020_

## Referee Comment (RC1) · Stefan Hergarten (Referee) · 15 Jul 2020

In this manuscript, analyses of topography and measurements of rock strength are used to unravel the deformation pattern of the Lesser Himalayan Duplex with emphasis on rapid exhumation during the last 2-3 Myr.

While I think that this study might contribute to the solution of an important research question, I must admit that I am disappointed by the manuscript. In its present form, its quality is far off from what I expect from this team of authors. It the first author was a

student, I would ask the supervisor to take a look at it before submitting. However, the first author is an experienced researcher who has published in highly ranked journals.

The problems concerning the quality of the manuscript occur at all levels. The structure is not really good at some points, e.g., parts of the discussion section read more like parts of an introduction, but the introductory part is already quite long. In general, too much of the reasoning relies son supplementary figures.

The applied morphometric methods appear to be appropriate in total, but the description does not allow for a serious assessment whether the choice of the parameters is good or not. Bumps in the order of magnitude of 100 m in the river profile shown in Fig. 2 do not increase my trust in the analysis. Some information is missing, e.g., the width of the swaths. Beyond this, the different metrics (channel steepness, specific stream power, channel width) are a bit isolated, and following the reasoning how they are combined to the whole picture is not easy.

Beyond these aspects, there are also technical issues that should have been detected by the coauthors. There are several errors in the text (not only typos), no all curves shown in Fig. 3 are explained, the two stretches shown in Fig. 2 b and c are not consistent with the marked regions in Fig. 2 a, and even the first reference I checked randomly (Hack 1973) does nor occur in the reference list.

In sum, I have no doubt that the work behind this manuscript is good, but the manuscript itself is not ready for being published in a scientific journal in its present state. From my point of view, a thorough rewriting is necessary.

---

## Referee Comment (RC2) · Adam Forte (Referee) · 15 Jul 2020

In their paper "Structural variations in basal decollement and internal deformation of the Lesser Himalayan Duplex trigger landscape morphology in the NW Himalayan interiors" authors Dey and others incorporate field observations, measurements of rock strength, and topographic analysis to try to better understand the underlying geometry of the Himalayan sole thrust and the patterns of active uplift in a specific part of the Lesser Himalayan Duplex. The paper is interesting, but it is missing a lot of fundamental detail and it's hard to follow the logic all the way to the conclusion. In my detailed

comments that follow, I try to highlight various portions of the manuscript that need a bit more description or justification. Many of these potentially rise to the level of 'Major Issue', but before I get to these, I highlight what I consider to maybe the largest issue with the manuscript in its current form.

Specifically, at present, it is challenging to see exactly how they arrive at their major conclusion (i.e. the added structural complexity within the sole thrust and the presence of two discrete ramps and flats). For example, in their summation figure 4, they highlight two pairs of low relief and high relief areas, which they relate to the underlying fault geometry corresponding to flats and ramps, respectively. However, in Figure 2 where they are showing the river data, it's not clear where this middle low-relief area is. The high and low relief patterns are similarly not particularly clear in Figure 3. Looking to the supplement (Fig S3), I similarly can't really figure out where the middle low relief area is supposed to be. Does this show up in other metrics (e.g. local relief) etc? Maybe more importantly, the direct jump from these patterns to the hypothesized structure is a bit abrupt. What observations are there to reject active surface breaking faults (as has been proposed by some authors mentioned in this paper)? Are the authors actually rejecting surface breaking faults (i.e. they show what might be one in Figure 4, but it's unclear whether they consider this an active portion of ramp 1 or a passive, dead thrust now be translated on the southern flat)? The main text talks only about ramps but the caption for figure 4 says that maybe there is surface breaking fault? How consistent is this other, more general studies regarding the topography over growing duplexes or movement over flats (references are presented in my detailed comments) Much of the uncertainty and ambiguity in the way data was collected or analyzed and choices made that I highlight in my more specific comments kind of feed into the uncertainty with regards to the final result, but even if all those are addressed, there still needs to be more connection drawn out between the observations they present and the model they propose. Ultimately, this could be a strong contribution showing how topographic observations and field observations can be incorporated to tell us something about deeper structure of orogenic systems, but it's not quite there at present. I hope that my

**ESurfD**
comments are useful to the authors to help them improve their manuscript.

Line-by-line comments:

L35-47 – Without some sort of figure, this opening assumes a fair bit of knowledge on the part of the reader on the location and geometry of the major Himalayan structures. While many are passingly familiar with these, it might be advisable to include a simple cartoon illustrating these structures and there general location with respect to topography. Maybe add a panel to figure 1 that accomplishes this? At least referencing figure 1 as is here could help, but not all of these structures are on here (MFT or STDS) and there are additional structures on Figure 1. Since much of the rest of paper hinges on which structures are active or not, not knowing where they are is kind of a detriment. Especially putting the LHD into structural context with these other structures seems crucial (and again, this can all be in a cartoon, not asking for a balanced cross section or anything). Other worthwhile question to consider which perhaps can put this issue in context, if this was any almost any other mountain range, would it be reasonable to expect a reader to know the relationships between the major local faults in an article published in a widely read, general geology journal like ESurf? You could cite fig S1 / S2 here, but I would argue that the knowledge of this information is sufficiently important to the main point of the paper that such a figure should be in the main text.

L216-217 – You need to explain a little bit more about how you're doing your basin wide statistics. It's not clear from this description, and the representation of it in Figure 3 is confusing (i.e. where are the basin boundaries, etc?).

L246-248 – RE the specific stream power calculation (1) you should state here in your methods that you assume constant discharge (not relegate it to the caption of Table 1) and perhaps more importantly (2) you need to justify that a constant discharge is applicable here. On figure 3, this is traversing >160 km of river distance and potentially traversing some large gradients in precipitation. As a worst case, you should be able to use the available TRMM data for the region to estimate discharge (this is simple with

TopoToolbox that you are already using as you can calculate flow accumulation with a precipitation raster is an optional input). Is there any discharge data in the region to compare this to? Perhaps you have good reason to assume constant discharge, but until that is shown in the paper, it's hard to know how to interpret the SSP data (or whether it should be believed at all).

L256-265 – Part 1 : Your measurements are your measurements, so you'll need to make do with what you've got, but it is worth discussing/addressing why you only collected 8-10 per location as this is ∼1/2 to 1/3 of the number of measurements thought to be needed to be robust (e.g. Niedzielski, T., Migon, P., Placek, A., 2009. A minimum sample size required from Schmidt hammer measurements. Earth Surface Processes and Landforms 34, 1713–1725. https://doi.org/10.1002/esp.1851). Perhaps it would be worth while considering pooling results from units/lithologies you consider to be similar and to get a larger N and seeing how those compare to the small N individual sites (i.e. if you have 5 sites all in the same unit with 10 measurements each, look at the statistics of the aggregate population of 50 values and see how those compare to the statistics of each of the 5 measurement sites). Such an analysis might help to alleviate some concerns, but there will remain lingering issues with a small N for each site. Similarly reporting of the raw values in a supplemental table and considering the standard deviation on the means when you're using them would be warranted (i.e. are the apparent differences in mean rebound values in Figure 3 'real'? how much of those differences would disappear or be less extreme if you considered the uncertainty?). Whether you follow my specific recommendations here or not, there needs to be more transparency with regards to these values and how reliable they may (or may not) be.

L256-265 – Part 2 : In general, a lot more detail is required to interpret your Schmidt hammer data. Later you describe significant fabrics in the rocks in the field (plus some nice field photos in the supplement). How did you consider this when taking your Schmidt hammer readings? Were you consistent with taking readings parallel or perpendicular to fabrics? Generally, where did you take these readings? Were

they in the active channels? On the banks? Were the measured faces wet (which can bias readings)? Did you evenly space your measurements? Did you avoid fractures (which can bias readings)? Did you try to measure near fractures (which can bias readings)? There is a rich literature on concerns related to Schmidt hammer readings, like the Niedzielski paper above, but Aydin, A., Basu, A., 2005. The Schmidt hammer in rock material characterization. Engineering Geology 81, 1–14. https://doi.org/10.1016/j.enggeo.2005.06.006 is a good source for why knowing the answers to at least some of the questions posed above are relevant to interpreting your results.

L319-322 – The knickpoints might be easier to visualize if you included a chi (integral of drainage area) vs elevation plot as a companion to the long-profile.

L338-339 – More discussion / consideration of the influence of dams and reservoirs on your width measurements might be warranted in your methods (it wasn't clear until now that there were dams on the profile).

L342-343 – Similar to the comments earlier with regards to the SSP calculation, it is worth considering how the widths you measure compare to drainage area / discharge. This also gets to the constant discharge assumption, i.e. what is the change in drainage area along the portion of the river you're examining? Obviously your width measurements are not varying smoothly as a function of drainage area, but this is an important contributor to channel width that appears to be largely ignored.

L354 – But assuming a constant discharge right? That's what Table 1 indicates.

L384- An alternative / complimentary approach might be thinking about patterns in cosmogenic erosion rates with topography. At least based on a quick browsing of the OCTOPUS database (https://earth.uow.edu.au/) there are no cosmo basins directly in your area, but there are some not that far away (Olen et al, 2016, Munack, 2014, Dortch 2011). There is good evidence of relationships between erosion rates and ksn (e.g. Kirby, E., Whipple, K.X., 2012. Expression of active tectonics in erosional

landscapes. Journal of Structural Geology 44, 54–75.) so you could explore what an aggregate of ksn vs E data in the surrounding regions imply for your area (need to consider complicating factors like precip and rock type when transporting relationships and restrict your analysis to locally equilibriated basins, but maybe at least another option you could consider). These would also be more on a complimentary timescale compared to thermochron, which might be giving you a longer term average.

L452 – Are these cooling ages on any of your figures / maps? It might help for spatial context. Could probably add them to Figure 4 without cluttering too much.

L492 – Worth considering how consistent your observations are with other studies focused on the surface / geomorphic expression of a growing duplex. The paper from Adams et al, 2016 (Adams, B.A., Whipple, K.X., Hodges, K.V., Heimsath, A.M., 2016. In situ development of high-elevation, low-relief landscapes via duplex deformation in the Eastern Himalayan hinterland, Bhutan. Journal of Geophysical Research: Earth Surface 121, 294–319. https://doi.org/10.1002/2015JF003508) might be a relevant one to consider. Similarly, you are ultimately arguing for motion over a series of ramps and flats. You may want to think about the role that lateral advection could play in the observed topographic patterns (e.g. Eizenhöfer, P.R., McQuarrie, N., Shelef, E., Ehlers, T.A., 2019. Landscape Response to Lateral Advection in Convergent Orogens Over Geologic Time Scales. J. Geophys. Res. Earth Surf. 124, 2056–2078. https://doi.org/10.1029/2019JF005100). In general, need to think / talk much more about how your observations lead to the model you propose, because at present, this is not clear at all.

Figure 2 – In a, what is the jagged blue line floating above the profile? Is it fluvial relief? A blow up the profile? There's no mention of it in the caption. In general, applying some amount of smoothing to the profile (and its derivatives) would be appropriate as it is hard to see the signals you're trying to highlight with the noise from the DEM superimposed.

**ESurfD**

Interactive
comment

---

## Short Comment (SC1) · 15 Jul 2020

Revered editor Prof. Gloaguen,

I understand your concern and will abide by that. Actually these two figures were readily available with me on which Dr. Hergarten focused on. His comments are helpful and we will start working on the text to make it more pointed and enjoyable.

Thanks, Saptarshi

[Figure]

**ESurfD**

Interactive
comment

---

## Editor Comment (EC1) · Richard Gloaguen (Editor) · 15 Jul 2020

Dear authors

I would strongly recommend to take your time before you answer to the reviewers. Make sure that the answers are useful for the review process and that you ponder the meanings of the constructive arguments.

---

## Author Comment (AC1) · 15 Jul 2020

Dear Dr. Hergarten,

First of all, thank you for your valuable feedback. I read your comments thoroughly and would like to raise a few points in this regard.

1. In Fig.2a, the river profile is bumpy and I agree to that. In those cases, the river width is less than 30m, therefore, SRTM is of no use. We had to go for ALOS PALSAR 12.5m DEM. still, with such narrow gorge around, picking the river-line is a bit difficult. That's

why the long profile is bumpy. But, for the analysis, we used the profile smoothing tool from topotoolbox, so, the bumpiness is partially nullified. For reference, I have added the smoothed long profile here. Please note that the ruggedness of data upstream from K2 is probably due to existence of a reservoir. The ruggedness remained even after using the hydrological fill function.

2. Swath width is mentioned in the figure caption, but we will add it in the figure in revision.

3. In case of figure 2c, the profile is 'mismatch' with the stretch shown in Fig. 2a. This is because, Fig.2c is an extrapolated long-profile. The Chenab river has a N-S traverse over the MCR-2. So, we took the upstream and downstream segment of the MCR-2 and projected it on a perpendicular traverse to the strike of the orogen/ the regional structures. so, the dx is fig.2c, corresponds to the width of the second crustal ramp and not the original along-river length. I have provided a sketch for the same.

4. Regarding the typos, We used the popular software named 'Grammarly' for thorough check of the text. I will re-run the check and revise it accordingly. On similar note, we will look at the text for a proper ordering and revise it soon.

Thanks again for such a quick review.
* * *
**Projection used for steep segment identification**

width of ramp

upstream stretch

line perpendicular to the
orientation of the ramp

downstream stretch

dx

**Fig. 1.** Illustration explaining orthogonal projection used in Fig. 2c

**Fig. 2.** revised smoothed long profile for Fig.2

---

## Author Response (AR1)

To

The Editor

Earth Surface Dynamics

Date – 14.09.2020

[Sub: submission of revised manuscript #esurf-2020-37 and detailed comments on reviews]

Respected Prof. Glouagen,

First of all, I would like to convey our heart-felt thanks to you and the two reviewers Prof. S. Hergarten and Prof. A. Forte for their insightful reviews. The reviews have immensely helped us to improve the scientific quality and presentation of the manuscript. I further thank you for allowing additional time for submission of the revised manuscript considering my physical condition.

The revised manuscript has undergone major changes based on the suggestions of the reviewers. We have re-analyzed a significant part of the terrain morphometry and shifted a handful of field data/photographs to the main manuscript from the supplement. But the most important change in the revised manuscript is the incorporation of sediment chronology. Since the first submission, we obtained a handful of depositional ages of sediments from the Kishtwar valley using luminescence dating method. Luminescence dating, with its uncertainties and limitations, has long been used to determine sediment burial ages. We used OSL dating of quartz and IRSL dating of K-feldspars to obtain the chronology of sediment aggradation in the Kishtwar region. The ongoing fluvial incision post-dates the largest valley aggradation recorded in this region, therefore, we are able to constrain the upper age limit of fluvial bedrock incision. Bedrock incision rates since late Pleistocene are in agreement with long-term exhumation rates from the study area confirming protracted growth of the Himalayan interiors from million-year to millennial timescales. We affirm that the incorporation of bedrock incision rates do not alter our initial research finding, but strengthen our study with evidence of orogenic growth over geomorphic/millennial timescale.

Addition of field photographs on structural styles, geomorphic features and sediment archives, as well as incorporation of sediment chronology and incision rate estimates have resulted in an increase in no. of figures from 4 in the initial manuscript to 10 in the revised version. One table containing the details of luminescence dating samples is also added in the revised manuscript.

We revised the manuscript title to 'Growth of the Lesser Himalayan Duplex and rapid fluvial incision modulate landscape morphology in tectonically-active Kishtwar Window in NW Himalayan interiors'.

With the addition of luminescence chronology in this manuscript, I would like to add Dr. Naveen Chauhan from PRL Ahmedabad as a co-author. He has been involved in luminescence dating along with me. The detailed list of changes and answers to the review comments are attached herewith.

Thanks again for considering our manuscript.

On behalf of the authors,

Saptarshi Dey

**Answers to the review comments:**

*Reviewer 1 (Stefan Hergarten):*

(Answers in blue font)

The problems concerning the quality of the manuscript occur at all levels. The structure is not really good at some points, e.g., parts of the discussion section read more like parts of an introduction, but the introductory part is already quite long. In general, too much of the reasoning relies on supplementary figures.

Thanks for your comments on the organization and arrangement of the manuscript. We also felt the same that parts of the manuscript is indeed a 'long read'. We revised and rechecked the possibilities of rearranging the text in introduction and discussion section. We reduced the redundant texts across these two sections and shortened the introduction. We are able to reduce some of the discussion part, which we believe we touched upon in the introduction. The supplementary figures are now incorporated in the main manuscript. They are largely clustered into 3 figures: - Figure 2 (structural styles and lithology), Figure 3 (Geomorphic features) and Figure 4 (Sediment archive stored along Chenab River).

The applied morphometric methods appear to be appropriate in total, but the description does not allow for a serious assessment whether the choice of the parameters is good or not. Bumps in the order of magnitude of 100 m in the river profile shown in Fig. 2 do not increase my trust in the analysis. Some information is missing, e.g., the width of the swaths. Beyond this, the different metrics (channel steepness, specific stream power, channel width) are a bit isolated, and following the reasoning how they are combined to the whole picture is not easy.

The morphometric analysis has been redone. The bumps in the river profile have been reduced and mostly removed by using smoothing with 20-pixel smoothing window (mentioned in the text). The swath widths are mentioned (3 km for stretch 1 and 2). All the morphometric analyses are now drafted in a single frame having same x-axis showing along-river distance (Fig.7). We hope that this revision will help us to visualize the variations in morphometric parameters and their significance in a much better way.

Beyond these aspects, there are also technical issues that should have been detected by the coauthors. There are several errors in the text (not only typos).

We rechecked the manuscript for typos and language/logical errors. We revised the entire text. The revised manuscript has been checked by a couple of skilled persons having higher proficiency in English language. We also had a thorough check with 'Grammarly'.

not all curves shown in Fig. 3 are explained.

The unexplained curves shown in figure 3 of the initial submission are now removed from the manuscript. In lieu of that, we provided field photos as a reference of changing valley morphology (Fig.3).

the two stretches shown in Fig. 2 b and c are not consistent with the marked regions in Fig. 2 a.

The two stretches are now marked in Fig.8a and the long-profiles are listed in Fig.8c and 8d. We guess the confusion is regarding the length and width of the stretch 2. All of us have to recognize that the stretch 2 (N-S traverse of the River) is basically parallel to the orogen or the proposed ramp structure. That is not the case with stretch 1. Stretch 1 is nearly perpendicular to the proposed ramp on which the duplex is growing. To estimate the orogen-perpendicular offset across the ramp, we used 'orthogonal projection of the orogen-perpendicular traverses of the River that can be seen on either side of the N-S traverse. That reasoning is now mentioned in the text in greater detail. For getting an idea of orogen-perpendicular width of the ramp (the steep segment), please refer to Fig.8a.

Even the first reference I checked randomly (Hack 1973) does not occur in the reference list.

We are sorry for the mistake. The reference list is now updated and thoroughly checked multiple times. In revision, a significant amount of vital references are added and some are removed.

From my point of view, a thorough rewriting is necessary.

We agree to your kind review comments and revised the whole text and the figures. In addition to the previously-shown morphometric analysis, in this revised version we have supplied a thorough understanding of the sediment aggradation in the valley and quantified bedrock incision rates using the sediment chronology based on luminescence dating method. This is the first attempt for assessment of late Pleistocene sediment deposition history in the entire middle Chenab valley. This addition helps us to compare long-term exhumation rates with millennial-scale fluvial incision rates and acts as a stand-alone metric for assessing the growth of the orogen-interior.

We would also like to refer you to go through some of the review comment answers for the second reviewer Prof. A. Forte. There are some more clarifications on the pertinent issues raised during review process.

***Reviewer 2 (Adam Forte):***

Line-by-line comments:

(Answers in blue font)

L35-47 – Without some sort of figure, this opening assumes a fair bit of knowledge on the part of the reader on the location and geometry of the major Himalayan structures. While many are passingly familiar with these, it might be advisable to include a simple cartoon illustrating these structures and there general location with respect to topography. Maybe add a panel to figure 1 that accomplishes this? At least referencing figure1 as is here could help, but not all of these structures are on here (MFT or STDS) and there are additional structures on Figure 1. Since much of the rest of paper hinges on which structures are active or not, not knowing where they are is kind of a detriment. Especially putting the LHD into structural context with these other structures seems crucial (and again, this can all be in a cartoon, not asking for a balanced cross section or anything). Other worthwhile question to consider which perhaps can put this issue in context, if this was any almost any other mountain range, would it be reasonable to expect a reader to know the relationships between the major local faults in an article published in a widely read, general geology journal like ESurf? You could cite fig S1 / S2 here, but I would argue that the knowledge of this information is sufficiently important to the main point of the paper that such a figure should be in the main text.

Thank you for pointing this out. At the first go, we were hesitant to put a conceptual sketch as Fig.1b. Now, with your comment, we added a general sketch of cross-section of NW Himalaya as Fig.1b. A detailed balanced cross-section is also there in the supplement S2. The overview of Himalayan orogen is drafted in supplement Fig.S1.

L216-217 – You need to explain a little bit more about how you're doing your basin wide statistics. It's not clear from this description, and the representation of it in Figure 3 is confusing (i.e. where are the basin boundaries, etc?).

More details of basinwide calculations are now added in line # 274-279. The whole analysis has been redone. We preferred to show the variation in elevation, topographic relief, annual rainfall data and basin wide mean ksn for the entire Chenab drainage (area shown in Fig.1a). That is drafted in Fig.5. The basin boundaries are also plotted. We hope that the figure is clear enough to distinguish the variations in regional scale as well as in smaller segments such as the KW.

L246-248 – RE the specific stream power calculation (1) you should state here in your methods that you assume constant discharge (not relegate it to the caption of Table 1) and perhaps more importantly (2) you need to justify that a constant discharge is applicable here. On figure 3, this is traversing >160 km of river distance and potentially traversing some large gradients in precipitation. As a worst case, you should be able to use the available TRMM data for the region to estimate discharge (this is simple with TopoToolbox that you are already using as you can calculate flow accumulation with a precipitation raster is an optional input). Is there any discharge data in the region to compare this to? Perhaps you have good reason to assume constant discharge, but until that is shown in the paper, it's hard to know how to interpret the SSP data (or whether it should be believed at all).

We agree to your concern and now added the statement that we used uniform discharge for SSP calculation in the main text (line #370-373). The reasoning behind assuming a constant discharge has been derived by looking at the TRMM data (as suggested by you). Please refer to Fig.5c and 5e. The interior of the Jammu-Kashmir Himalaya has no orographic barrier and receives more-or-less uniform rainfall <1.5 m/y. That is why we had put the term 'discharge-normalized' in the first place.

L256-265 – Part 1 : Your measurements are your measurements, so you'll need to make do with what you've got, but it is worth discussing/addressing why you only collected 8-10 per location as this is ~1/2 to 1/3 of the number of measurements thought to be needed to be robust (e.g. Niedzielski, T., Migon, P., Placek, A., 2009. A minimum sample size required from Schmidt hammer measurements. Earth Surface Processes and Landforms 34, 1713–1725. https://doi.org/10.1002/esp.1851). Perhaps it would be worthwhile considering pooling results from units/lithologies you consider to be similar and to get a larger N and seeing how those compare to the small N individual sites (i.e. if you have 5 sites all in the same unit with 10 measurements each, look at the statistics of the aggregate population of 50 values and see how those compare to the statistics of each of the 5 measurement sites). Such an analysis might help to alleviate some concerns, but there will remain lingering issues with a small N for each site. Similarly reporting of the raw values in a supplemental table and considering the standard deviation on the means when you're using them would be warranted (i.e. are the apparent differences in mean rebound values in Figure 3 'real'? how much of those differences would disappear or be less extreme if you considered the uncertainty?). Whether you follow my specific recommendations here or not, there needs to be more transparency with regards to these values and how reliable they may (or may not) be.

Thanks for recommending the nice article on Schmidt hammer experiment. We agree to the fact that our measurement-per site is lower than what would be statistically-meaningful. That is partly due to the awful fact that I have not been allowed by the army to have a pit-stop for more than 10-15 minutes at a single site. Neither was I allowed to roam off-road. In that stipulated time, I had to do whatever I could, including observation, photography, sample procurement and taking hardness measurements. Thankfully, they allowed more time where I took luminescence samples. I understand that this reasoning sounds weak and weird, but that much difficulty is expected in Kashmir.

Therefore, I followed your suggestion of pooling all the results from same litho-units together and calculated the mean ± 1σ for that population. Raw measurements are listed in Supplementary Table ST1. I have removed the data taken close to fractures and also removed the data from the tributary

Maru River. These steps resulted in reduction in total no. of sites from 96 to 75 as we removed biased data. The recalculated R values are plotted along the long-profile in Fig.7e.

L256-265 – Part 2: In general, a lot more detail is required to interpret your Schmidt hammer data. Later you describe significant fabrics in the rocks in the field (plus some nice field photos in the supplement). How did you consider this when taking your Schmidt hammer readings? Were you consistent with taking readings parallel or perpendicular to fabrics? Generally, where did you take these readings? Were they in the active channels? On the banks? Were the measured faces wet (which can bias readings)? Did you evenly space your measurements? Did you avoid fractures (which can bias readings)? Did you try to measure near fractures (which can bias readings)? There is a rich literature on concerns related to Schmidt hammer readings, like the Niedzielski paper above, but Aydin, A., Basu, A., 2005. The Schmidt hammer in rock material characterization. Engineering Geology 81, 1–14. https://doi.org/10.1016/j.enggeo.2005.06.006 is a good source for why knowing the answers to at least some of the questions posed above are relevant to interpreting your results.

All the measurements were done perpendicular to the bedding/foliation planes of dry rocks along road-cut section. No data is taken from riverbanks. Sentences mentioning these facts are drafted in line #389-396. The R values reported in the revised version were taken away from visible fractures. We tested the hardness values of fractured surfaces; they are lower by ~20 units in R-value than non-fractured ones.

L319-322 – The knickpoints might be easier to visualize if you included a chi (integral of drainage area) vs elevation plot as a companion to the long-profile.

Post-smoothing of the long-profile using a 20-pixel moving window identifying the knickpoints was easier. To be sure, we ran the 'knickpointfinder' tool in Topotoolbox with a threshold 'dz' value of 30m. That confirmed our hand-picked knickpoints in Fig.7a. For your reference chi vs. elevation plot is drafted in supplement Fig.S4.

L338-339 – More discussion / consideration of the influence of dams and reservoirs on your width measurements might be warranted in your methods (it wasn't clear until now that there were dams on the profile).

We agree to the fact that the dams and reservoirs have altered the 'natural' channel width. Both the reservoirs expectedly have wider channels due to damming. One might argue that the dams were made on pre-existing geological structures which had gentler channel slopes in the upstream. But, we have very little or rather no access to the dam sites to confirm this. Considering the channel width downstream from the dams, there should exist a control of the water release from the dam. To eradicate that bias, we used the satellite image from summer monsoon months when these dams are open due to higher discharge caused by glacial melting and monsoon precipitation. Therefore, the channel width we showcase is expectedly the highest of the year.

However, there is more external control on channel width. In some parts, let's say, downstream from the point K2 and K3 (cf. Fig.7a and 7b), the channel width are lower because of formation of epigenetic gorge. The issue of epigenetic gorge formation is discussed in section 5.1.3. and 5.3.

L342-343 – Similar to the comments earlier with regards to the SSP calculation, it is worth considering how the widths you measure compare to drainage area / discharge. This also gets to the constant discharge assumption, i.e. what is the change in drainage area along the portion of the river you're examining? Obviously your width measurements are not varying smoothly as a function of drainage area, but this is an important contributor to channel width that appears to be largely ignored.

Agreed. But, if we look at the aspect of change in drainage area, the most obvious change in drainage area occurs when the tributary Maru R. joins the Chenab R. now, if we look in the channel width before and after the joining of the tributary, we see no significant change in channel width. In fact, that is our argument that there is some other (tectonic) control on channel width. Where, there is no tectonic control, let's say, from ~70km-15 km of the long-profile, we see monotonous increase in channel width with increasing drainage area.

L354 – But assuming a constant discharge right? That's what Table 1 indicates.

Revised accordingly.

L384- An alternative / complimentary approach might be thinking about patterns in cosmogenic erosion rates with topography. At least based on a quick browsing of the OCTOPUS database (https://earth.uow.edu.au/) there are no cosmo basins directly in your area, but there are some not that far away (Olen et al, 2016, Munack, 2014, Dortch 2011). There is good evidence of relationships between erosion rates and ksn (e.g. Kirby, E., Whipple, K.X., 2012. Expression of active tectonics in erosional landscapes. Journal of Structural Geology 44, 54–75.) so you could explore what an aggregate of ksn vs E data in the surrounding regions imply for your area (need to consider complicating factors like precip and rock type when transporting relationships and restrict your analysis to locally equilibriated basins, but maybe at least another option you could consider). These would also be more on a complimentary timescale compared to thermochron, which might be giving you a longer term average.

Stefanie in her paper (Olen et al., 2016) emphasized on the power-law scaling relationship between topographic metric and erosion rates. The power-law equation stands as-

$$Ksn = a * (erosion\ rate)^b$$

For the pan-Himalayan dataset, the paper predicts a = 215 and b= 0.3. If we incorporate the high ksn values (>500) obtained from KW and the surroundings, we obtain absurdly-high predicted erosion-rate for the study area (e ~ 15-16 mm/y). That tells us the fact that this power-law fit probably can't explain the true relationship of erosion vs. ksn values. Both, Rasmus and me are also preparing one manuscript on paleo erosion rates vs. topographic steepness in the nearby Dhauladhar Range. We hardly find the power-law scaling to be true there.

But, now we have an additional constraint on landscape change over geomorphic timescales. We obtained a handful of OSL-IRSL ages from aggraded sediment sequence stored above the bedrock straths in the Kishtwar valley. We demonstrate the sediment bodies in Fig.4, Fig.9. Details of sediment aggradation and their relationship with fluvial bedrock incision are drafted in section 3.3. methods of luminescence dating of transiently-stored sediments in and around Kishtwar, section 4.3.results of luminescence chronology, section 5.2. discussion on sediment aggradation in Chenab valley  and section 5.3. discussion on drainage reorganization and strath terrace formation along Chenab River. Based on the stratigraphic relationship of the deposits and the underlying bedrock straths, we calculated fluvial bedrock incision rates over millennial timescales. Our estimates are in agreement with long-term exhumation rates. But, as we are able to come up with bedrock incision rates since 80 ky, we believe that the issue of mismatch in timescale is addressed.

L452 – Are these cooling ages on any of your figures / maps? It might help for spatial context. Could probably add them to Figure 4 without cluttering too much.

AFT cooling ages (Kumar et al., 1995) are shown in Fig.1a in revised version.

L492 – Worth considering how consistent your observations are with other studies focused on the surface / geomorphic expression of a growing duplex. The paper from Adams et al, 2016 (Adams, B.A., Whipple, K.X., Hodges, K.V., Heimsath, A.M., 2016. In situ development of high-elevation, low-relief landscapes via duplex deformation in the Eastern Himalayan hinterland, Bhutan. Journal of Geophysical Research: Earth Surface 121, 294–319. https://doi.org/10.1002/2015JF003508) might be a relevant one to consider. Similarly, you are ultimately arguing for motion over a series of ramps and flats. You may want to think about the role that lateral advection could play in the observed topographic patterns (e.g. Eizenhöfer, P.R., McQuarrie, N., Shelef, E., Ehlers, T.A., 2019. Landscape Response to Lateral Advection in Convergent Orogens Over Geologic Time Scales. J. Geophys. Res. Earth Surf. 124, 2056–2078. https://doi.org/10.1029/2019JF005100). In general, need to think / talk much more about how your observations lead to the model you propose, because at present, this is not clear at all.

Thanks for recommending the paper by Adams et al., (2016). I went through it and on the basis of that, I can draw a few initial comments. Some of the findings of Adams et al. are in agreement with our data. We also observe a physiographic transition at the western margin of the KW, exposing the LH duplex. However, the low-relief landscape they propose to be sitting on a blind duplex at the stoss side of the transition, doesn't seem to fit in our context. The low-relief zone we have in the frontal Higher Himalaya show no sign of gorge formation and very limited fluvial bedrock incision. Therefore, we may argue that the duplex is limited within the KW. The duplex in KW is exposed to the surface and show structural variations (detected by changes in orientation). The observed steepening and narrowing of channel correlate well with segments of steep bedrock orientation. We argue that the steepening of river reflects the differential uplift along river triggered by existing ramp-flat geometry of the basal decollement. Unless the growth of the duplex is continuous, the topographic response would have been much subdued. For example, we may refer to the topographic metrics at the frontal MCT brittle fault zone. The MCT is no longer active, and that's why the morphometric parameters are subdued. To continue more on the paper by Adams et al., the structural architecture of the Bhutanese Himalaya is distinctly different from the NW Himalaya. That is why we compared our findings with studies from NW Himalaya (Nennewitz et al., 2018; Gavillot et al., 2018; Thiede et al., 2017).

In connection to your opinion on the role of lateral advection in topographic build-up, I would like to show the regional map of topographic relief and basinwide ksn distribution (Fig.5b,5d). In this study, we have no evidence of a lateral ramp in KW. The bedrock orientations shown in Fig. 8a attests to that. In the nearby Dhauladhar Range (DR) (SE of the KW), Thiede et al., (2017) proposed a frontal ramp beneath the Dhauladhar uplifting the topography. Now these two rapidly-uplifting zones have along-strike difference in the distance from the mountain front. Now referring to the topographic relief as well as the ksn map, we see that these two zones (KW and the Dhauladhar) are connected by a ~50-kn wide N-S trending zone of high relief and steepness. At this moment, we don't have structural data from this proposed link segment, but, by looking at the regional topographic pattern we would argue that, lateral ramp, if any, would be this one. But, at this moment, we have no data to comment anything on impact or evidence of lateral advection on KW.

[Figure]

Figure showing distribution of topographic relief in far western Himalaya. Quaternary exhumation rates are shown in grey and Pleistocene-Holocene shortening rates on active faults shown in purple-colored boxes. The proposed lateral ramp is shaded by transparent-red color.

Figure 2 – In a, what is the jagged blue line floating above the profile? Is it fluvial relief? A blow up the profile? There's no mention of it in the caption. In general, applying some amount of smoothing to the profile (and its derivatives) would be appropriate as it is hard to see the signals you're trying to highlight with the noise from the DEM superimposed.

The longitudinal profile has been revised. Smoothing has been performed with 20-pixel smoothing window. That removed the noise (Figure 6, 7a).

**Additions/ changes in the manuscript:**

Revised title:

Growth of the Lesser Himalayan Duplex and rapid fluvial incision modulate landscape morphology in tectonically-active Kishtwar Window in NW Himalayan interiors

Revised abstract:

The Kishtwar Window (KW) of the NW Himalaya exposes the north-western termination of the orogen-parallel anticlinal stack of thrust nappes, known as the Lesser Himalayan Duplex. However, its exact tectonic deformation pattern, geographic extent and activity are still debated. Here we combine morphometric analyses with structural data, field evidence and chronological constraints to describe the spatial pattern of internal deformation of the duplex. We agree with previous findings that the variations in the geometry of the basal décollement, the Main Himalayan Thrust (MHT) is important; however, the observed topography and neotectonic deformation can be explained only with additional internal faulting within the duplex. We recognize two significant steep stream segments/ knickzones, one in the center of the window, and a second one along its western margin, which we relate to fault-ramps emerging from the MHT. The larger of the knickzones corresponds to highly-fractured and folded rocks at the base of the steep stream segment suggesting internal deformation of the duplex, possibly linked to surface-breaking thrust fault-ramp at the core of the duplex. The second steepened knickzone coincides with the western margin of the window and is identified by a narrow channel through a comparatively weaker bedrock gorge. Luminescence dating of sediments overlying the bedrock strath provides the upper limit of terrace abandonment. We deduced a minimum of 3.1-3.5 mm/y fluvial bedrock incision on the MHT-fault ramp which is in overall agreement with long-term exhumation rates from the KW. Summarizing our findings, we favor a structural and active tectonic control on the growth of the duplex even over geomorphic timescales.

Line # 58-60: text replaced ‘’ with ‘over millennial timescales’.

Line # 77-78: added sentence- While most of the published cross-sections of the Himalayan orogen today recognize the duplex (Webb et al., 2011; Mitra et al., 2010; DeCelles et al., 2001), usually very little or no data is available whether the duplex is active over millennial timescales, and potentially a source of major Holocene earthquakes.

Line # 139-140: Added text- ‘4. Can we obtain new constraints on deformation over millennial timescales? Do millennial-scale fluvial incision rates support long-term exhumation rates?’

Line # 146: removed text- ‘’.

Line # 147-156: Added sentence- 'We used basinwide steepness indices and specific stream power calculation (derived from channel gradient and channel width) as a proxy of the fluvial incision. And, lastly but most importantly, we are able to constrain the fluvial bedrock incision rates by using depositional ages of aggraded sediments along Chenab River.'

Line # 160-162: Added sentence- 'Our new estimates on the bedrock incision rate agree with Quaternary exhumation rates from the KW, which mean consistent active growth of the duplex over million-year to millennial timescales.'

Line #159: Changed section header to –'2.Geological background and field observations'.

Line # 204-250: Sentences added on field observations with reference to field photographs in Fig.2, Fig.3 and Fig.4.

The Higher Himalayan sequence dips steeply away from the duplex (~65° towards west) (Fig.2a). The frontal horses of the LH duplex expose internally-folded greenschist facies rocks. Although at the western margin of the duplex, the quartzites stand sub-vertically (Fig.2b), the general dip amount reduces as we move from west to east for the next ~10-15 km up to the core of the KW. Near the core of the KW, we observed highly-deformed (folded and multiply-fractured) quartzite and granites at the core of the KW (Fig.2d, 2e). We also observed deformed quartz veins of at least two generations, as well as macroscopic white mica. Here, the river is also very steep and narrow; the rock units are also steeply-dipping towards the east (~55-65°) and are nearly isoclinal and strongly deformed at places (Fig.2f). Towards the eastern edge of the window, however, the quartzites dip much gently towards the east (~20-30°) and much lesser folding and faulting have been recognized in the field (Fig.2g).

The valley profile near the town of Padder at the eastern margin of the KW is U-shaped (Fig.3a). At the core of the KW, the Chenab River maintains narrow channel width and a steep gradient (Fig.3b). The E-W traverse of the Chenab River through the KW is devoid of any significant sediment storage. However, along the N-S traverse parallel to the western margin of the KW, beneath the Kishtwar surface, ~150-170m thick sedimentary deposits are transiently-stored over the steeply-dipping Higher Himalayan bedrock (Fig.3c). The height of the Kishtwar surface from the Chenab River is ~450m, which means ~280m of bedrock incision by the river since the formation of the Kishtwar surface. Along the N-S traverse of the River, epigenetic gorges are formed as a result of damming of paleo-channel by hillslope debris flow, followed by establishment of a newer channel path (Ouimet et al., 2008; Kothyari and Juyal, 2014). One example of such epigenetic gorge formation near the town of Drabshalla is shown in Fig.3d. Downstream from the town of Drabshalla, the River maintains narrow channel width (< 25 m) and flows through a gorge having sub-vertical valley-walls (Fig.3e). The tributaries originating in the Higher Himalayan domain form one major knickpoint close to the confluence with the trunk stream (Fig.3f). We have identified at least three strath surface levels above the present-day river channel, viz., T1 (280±5 m), T2 (170-175 m) and T3 (~120±5 m), respectively (Fig.3g). The first study on sediment aggradation in Middle Chenab valley (transect from Kishtwar to Doda town) was published by Norin (1926). He argued the sediment aggradation in and around the Kishtwar town is largely contributed by fluvioglacial sediments and the U-shaped valley morphology is a marker of past glacial occupancy. We partially agree to the findings of Norin (1926) and Ul Haq et al., (2019) as we observe ~100m thick fluvioglacial sediment cover unconformably overlying the Higher Himalayan bedrock (Fig.4b). The fluvioglacial sediments included alternate layers of pebble conglomerate and coarse-medium sand (Fig.4c). The pebbles are moderately rounded and polished suggesting significant fluvial transport. Our field observations suggest that the fluvioglacial sediments have been succeeded by a significant volume of hillslope debris (Fig.4c). The thickness of the debris-flow deposits is variable. The hillslope debris units contain mostly coarse-grained, highly-angular, poorly-sorted quartzite clasts from the frontal horses of the Lesser Himalayan Duplex. The hillslope debris units also contain a few fine-grain sediment layers trapped in between two coarse-grained debris layers (Fig.4e). The town of Kishtwar is situated on this debris flow deposit.

Line # 258-261: Added sentence- 'We compiled the topographic relief over a circular moving window of 4 km diameter (Fig.5b) and the rainfall distribution of the Chenab region (Fig.5c). The rainfall distribution is adapted from 12-year-averaged annual rainfall data from TRMM database (Bookhagen and Burbank, 2006).'

Line # 262-295: Text reorganized- section on Basin-wide normalized steepness indices revised as section 3.1.1.

Line # 274-285: Text added- 'Stream-specific ksn values in and around the KW are drafted in Supplementary Fig.S3. The catchments were delineated by using a maximum threshold of 200 sq. km, so that the basins we pick are smaller in size. The stream-specific $k_{sn}$ values were rasterized in ArcGIS and were extrapolated to the respective catchments using the zonal statistics toolbox. Basinwide mean $k_{sn}$ values for the delineated watersheds are portrayed in Fig.5d. Basinwide mean $k_{sn}$ values are plotted using a 500 $km^2$ threshold catchment area (Fig. 2d).

A 50-km-wide swath profile along line AB (cf.Fig.5a) show variation in elevation, mean annual rainfall and mean $k_{sn}$ values in the area (Fig.2e).'

Line # 279-283: Text removed- 'The slope breaks, known as the knickpoints (sometimes referred to as knickzones if it is manifested by a series of rapids instead of a single sharp break in profile), were allocated by comparing the change of slope along the distance-elevation plot (Fig.2a). The threshold 'dz' value (projected stream offset across a knickpoint) for this study is 30m.'

Line # 286-295: Text removed.

Line # 282: Change in section number to 3.1.2. Drainage network extraction.

Line # 286: Added text-'(250 m smoothing window)'.

Line # 305-319: Added text- 'Identification of the knickpoints/ knickzones and their relationship with the rock-types as well as with existing structures are necessary to understand the causal mechanism of the respective knickpoints/ knickzones. Knickpoints/(zones) can be generated by lithological, tectonic and structural control. Lithological knickpoints are stationary and anchored at the transition from the soft-to-hard substrate. The tectonic knickpoints originate at the active tectonic boundary and migrate upstream with time. Structural variations, such as ramp-flat geometry of any emerging thrust may cause a quasistatic knickpoint at the transition of the flat-to-ramp of the fault. In such cases, the ramp segment is characterized by higher steepness than the flat segment and often the ramp is characterized by a sequence of rapids, forming a wide knickzone, instead of a single knickpoint.

Longitudinal profile of the Chenab River show oversteepening across the KW (Fig.6), therefore, we focused on that segment (marked by red rectangle, cf. Fig.6) for further analysis. Longitudinal profile of the selected segment is shown in Fig.7a.'

Line # 339: Altered text- '50' replaced by '100'.

Line # 340: Altered text- 'Twenty' replaced by 'Ten'.

Line # 359-360: added sentence- 'Variations in channel gradient and ksn values along the longitudinal profile of the selected stretch are shown in Fig.7c and 7d, respectively.'

Line # 370-372: Added sentence- 'With the available TRMM data, we argue that the rainfall distribution in the study area is almost uniformly low (<1.5 m/y) (Fig.5c and 5e) and therefore we assumed a uniform discharge (Q) for SSP calculation.'

Line # 387: changed number- from '96' to '74'. (Removed data from tributary valleys, data from sites having irregular fractures)

Line # 389-396: Added sentences- 'All the measurements were taken perpendicular to the bedding/ foliation plane and no measurements are from wet surfaces or surfaces showing fractures. Each reading was taken at least 0.5m apart from the previous one. Average rock strength data collected from each of the test locations are plotted against the longitudinal river profile and channel width data in Fig.7e. Our data from individual sites are smaller in number than what is preferred for checking the statistical robustness of Schmidt hammer data (Niedzielski et al., 2009). Therefore, we combined the data from all sites representing similar lithology and portrayed the mean ±standard deviation for the same.'

Line # 397-465: Added method section on '**3.3. luminescence dating of transiently-stored sediments in and around Kishtwar**'

[revised manuscript text omitted]

Supplementary Table ST1:

| Site | latitude | longitude | rock type | Dist. (km) | R1 | R2 | R3 | R4 | R5 | R6 | R7 | R8 | R9 | R10 | R11 | R12 | Mean_R | stdev_R | lithology_mean R | lithology_R_sigma |
|---|---|---|---|---|---|---|---|---|---|---|---|---|---|---|---|---|---|---|---|---|
| 1 | 33.1842 | 75.3047 | LH MS | 1 | 39.9 | 39.3 | 40.6 | 40.3 | 42.3 | 41.5 | 40 | 40.7 | 40.9 | | | | 40.6 | 0.8 | 41.1 | 1.9 |
| 2 | 33.1664 | 75.3123 | LH MS | 2.3 | 40 | 40.2 | 40 | 39 | 39.2 | 40.3 | 39.2 | 40.9 | 41 | 41.2 | | | 40.0 | 0.7 | | |
| 3 | 33.1604 | 75.3253 | LH MS | 4.2 | 39.8 | 39 | 39.5 | 40.2 | 39.1 | 40.2 | 40.4 | 38.9 | 39.2 | 39.4 | 39.6 | | 39.6 | 0.5 | | |
| 4 | 33.1497 | 75.3485 | LH MS | 6.5 | 41.3 | 41.5 | 40.5 | 43.8 | 43.4 | 38.9 | 39.2 | 41.8 | 42 | | | | 41.4 | 1.6 | | |
| 5 | 33.139 | 75.3671 | LH MS | 10.8 | 40.1 | 37.8 | 37.7 | 37.2 | 39 | 38.8 | 39.2 | 39 | 39.1 | 38.5 | | | 38.7 | 0.9 | | |
| 6 | 33.1343 | 75.3797 | LH MS | 13.4 | 42 | 40.6 | 40.2 | 39.6 | 39.9 | 42.2 | 41.4 | 41.8 | 41.1 | 41.6 | | | 41.0 | 0.9 | | |
| 7 | 33.1317 | 75.4039 | LH MS | 17.5 | 43.1 | 43.5 | 40.2 | 40.9 | 43.9 | 45 | 44.1 | 43.4 | 42.9 | | | | 43.0 | 1.4 | | |
| 8 | 33.1266 | 75.417 | LH MS | 19 | 44 | 43.8 | 45.2 | 43.7 | 42 | 40.2 | 40.8 | 43.3 | 44.4 | 44.6 | | | 43.0 | 1.6 | | |
| 9 | 33.1287 | 75.4281 | LH MS | 22.1 | 43.1 | 43.2 | 44.5 | 44 | 44.7 | 40.3 | 41.6 | 42.3 | 43 | 42.9 | | | 43.0 | 1.3 | | |
| 10 | 33.134 | 75.4411 | Haimanta | 24.6 | 45 | 44.3 | 44.4 | 41.5 | 40.8 | 40 | 43.4 | 43.2 | 42 | 40.1 | 40.5 | 44 | 42.4 | 1.7 | 44.5 | 2.2 |
| 11 | 33.1306 | 75.4547 | Haimanta | 26.2 | 47.2 | 47 | 45.5 | 48.2 | 47.9 | 46.6 | 47 | 47.1 | | | | | 47.1 | 0.8 | | |
| 12 | 33.1373 | 75.4619 | Haimanta | 28.8 | 47.7 | 48 | 49.1 | 48.2 | 49 | 46.2 | 46.6 | 46.7 | 47.3 | 47.8 | | | 47.7 | 0.9 | | |
| 13 | 33.1372 | 75.4747 | Haimanta | 32.1 | 43.4 | 42 | 42.9 | 44.1 | 42 | 43.1 | 43.6 | 43.3 | 43.2 | 43 | 41 | | 42.9 | 0.8 | | |
| 14 | 33.1352 | 75.4843 | Haimanta | 33.2 | 43.5 | 43.7 | 44 | 43.3 | 43.3 | 44.4 | 44.5 | 44 | 42.8 | 43.9 | | | 43.7 | 0.5 | | |
| 15 | 33.1398 | 75.4995 | Haimanta | 34 | 44.1 | 44.5 | 45 | 44.5 | 44.3 | 44.8 | 43.6 | 43.8 | 44.4 | | | | 44.3 | 0.4 | | |
| 16 | 33.1404 | 75.5219 | HHCS | 36.9 | 36.6 | 36.9 | 35.3 | 35.9 | 36.6 | 37 | 37.3 | 36.8 | 38 | 37.1 | 35.6 | 36 | 36.6 | 0.7 | 38.9 | 2.4 |
| 17 | 33.1316 | 75.5482 | HHCS | 38.1 | 41.1 | 41.2 | 42.6 | 40 | 40.4 | 41.3 | 42.8 | 38 | 42.3 | 38.7 | 40.9 | | 40.8 | 1.4 | | |
| 18 | 33.1362 | 75.5618 | HHCS | 42 | 40.8 | 40.3 | 42.3 | 39 | 39.4 | 40.9 | 42.2 | 42 | 39.6 | 40 | 40.1 | | 40.6 | 1.1 | | |

| No | Lat | Long | Type | D | m1 | m2 | m3 | m4 | m5 | m6 | m7 | m8 | m9 | m10 | m11 | m12 | Mean | SD | | |
|---|---|---|---|---|---|---|---|---|---|---|---|---|---|---|---|---|---|---|---|---|
| 19 | 33.1361 | 75.5857 | HHCS | 43.6 | 38.8 | 39 | 39.2 | 39.9 | 37.2 | 37.9 | 39.9 | 38.3 | 38 | 37.9 | 38 | | 38.6 | 0.8 | | |
| 20 | 33.132 | 75.6025 | HHCS | 45 | 35.9 | 36 | 36.2 | 35.8 | 37.2 | 36.9 | 38 | 36.8 | 39 | 37.5 | 37.2 | 38.2 | 37.1 | 1.0 | | |
| 21 | 33.1211 | 75.6232 | HHCS | 47.5 | 39 | 39.8 | 39 | 38.9 | 37.1 | 40.4 | 40.1 | 37.3 | 38 | 36.9 | 40.3 | | 38.8 | 1.2 | | |
| 22 | 33.1217 | 75.6399 | HHCS | 48.2 | 39.5 | 38.9 | 39 | 39.8 | 40.2 | 37.1 | 40.4 | 38.4 | 37.9 | 37 | 38.1 | 37.5 | 38.7 | 1.1 | | |
| 23 | 33.1304 | 75.656 | HHCS | 48.7 | 42.1 | 40 | 39.8 | 37.5 | 37.8 | 37.8 | 38.6 | 39 | 38.4 | 37.3 | 39.9 | | 38.9 | 1.4 | | |
| 24 | 33.1498 | 75.6753 | HHCS | 49.3 | 40.4 | 40.2 | 41.1 | 41.3 | 45 | 40.7 | 38.4 | 40.2 | 43.7 | 44 | 37.6 | | 41.1 | 2.2 | | |
| 25 | 33.1471 | 75.6649 | HHCS | 50 | 37.7 | 38 | 38.9 | 37.8 | 39.2 | 39.1 | 38.8 | 39 | 39.9 | 37.7 | | | 38.6 | 0.7 | | |
| 26 | 33.1489 | 75.7136 | HHCS | 51.1 | 40.7 | 40.7 | 40 | 38.8 | 39.1 | 39.1 | 39.7 | 39.2 | 40 | 37.8 | 37.4 | 37.3 | 39.2 | 1.1 | | |
| 27 | 33.1527 | 75.7408 | HHCS | 51.4 | 40.5 | 40.3 | 39.8 | 36.9 | 37 | 42.3 | 40.3 | 40 | 37.2 | 38.7 | 39 | | 39.3 | 1.6 | | |
| 28 | 33.1352 | 75.759 | HHCS | 53 | 42.4 | 42.2 | 40.6 | 45.7 | 40.2 | 39.2 | 39.9 | 42.1 | 43 | 43.2 | 41.5 | | 41.8 | 1.7 | | |
| 29 | 33.149 | 75.8031 | HHCS | 55.2 | 39.6 | 39.9 | 39.2 | 39 | 38.3 | 40.6 | 41.1 | 39.9 | 38.6 | 39.8 | 40 | | 39.6 | 0.8 | | |
| 30 | 33.1611 | 75.8064 | HHCS | 57.5 | 38 | 36.8 | 37 | 37.2 | 38.9 | 36.6 | 36.4 | 37.2 | 37 | 38.2 | 38.1 | | 37.4 | 0.7 | | |
| 31 | 33.1832 | 75.8146 | HHCS | 60.3 | 37.6 | 36.9 | 37 | 35.8 | 36 | 37.2 | 38.1 | 36.7 | 35.4 | 36.7 | | | 36.7 | 0.8 | | |
| 32 | 33.1913 | 75.8107 | HHCS | 63.4 | 36 | 34.4 | 35.8 | 32.3 | 35 | 30.1 | 28.9 | 33.1 | 32.2 | 31.5 | 35.3 | | 33.1 | 2.3 | | |
| 33 | 33.2068 | 75.8036 | HHCS | 67.9 | 39.3 | 39.2 | 37.9 | 37.3 | 40 | 38.4 | 38.6 | 39.8 | 40.1 | 39.6 | 39.9 | | 39.1 | 0.9 | | |
| 34 | 33.2256 | 75.8014 | HHCS | 72.2 | 39.9 | 40.6 | 40.3 | 39.8 | 39.8 | 41.3 | 41.5 | 40.8 | 41 | 41.1 | 41.2 | | 40.7 | 0.6 | | |
| 35 | 33.2304 | 75.7918 | HHCS | 73.6 | 44.7 | 40.1 | 40.2 | 40.2 | 41.1 | 41.3 | 40.4 | 43.2 | 42.6 | 40.4 | 40.9 | | 41.4 | 1.4 | | |
| 36 | 33.2405 | 75.7895 | HH mg | 74.5 | 37.7 | 34.2 | 30.4 | 30.6 | 32.2 | 30.9 | 31.1 | 30 | 34.1 | 30.4 | | | 32.2 | 2.3 | 48.9 | 6.7 |
| 37 | 33.2493 | 75.7824 | HH mg | 75.4 | 44 | 44.9 | 43.9 | 43.8 | 44.2 | 45.7 | 45.2 | 43.1 | 43 | 42.6 | 43 | 44 | 43.9 | 1.0 | | |
| 38 | 33.2614 | 75.7753 | HH mg | 76.8 | 43.7 | 44 | 45.3 | 45.2 | 44.8 | 44.3 | 44.9 | 45 | 45.5 | 44.8 | 43.1 | 45.2 | 44.6 | 0.7 | | |
| 39 | 33.2674 | 75.7746 | HH mg | 77.3 | 45.5 | 45.3 | 46.2 | 45.2 | 44.9 | 45.3 | 48 | 46.2 | 46.3 | 46.8 | | | 46.0 | 0.9 | | |
| 40 | 33.2728 | 75.7698 | HH mg | 78.8 | 45.3 | 45.4 | 47 | 46.7 | 45.9 | 45.3 | 46 | 45.6 | 45.1 | 45 | 46 | | 45.8 | 0.6 | | |

| # | Lat | Long | Type | Dist | m1 | m2 | m3 | m4 | m5 | m6 | m7 | m8 | m9 | m10 | m11 | Mean | SD | | |
|---|---|---|---|---|---|---|---|---|---|---|---|---|---|---|---|---|---|---|---|
| 41 | 33.2829 | 75.7699 | HH mg | 79.2 | 48 | 46.6 | 45.9 | 46 | 45.7 | 46 | 45.8 | 46.3 | 45.9 | 46.3 | 46 | 46.2 | 0.6 | | |
| 42 | 33.2897 | 75.7644 | HH mg | 80.1 | 45.9 | 47.8 | 48 | 49.2 | 49.4 | 48.5 | 49.5 | 47.3 | 48.9 | 49.5 | 49.2 | 48.5 | 1.1 | | |
| 43 | 33.2918 | 75.754 | HH mg | 80.6 | 46.9 | 46.5 | 47.2 | 47.2 | 46.9 | 47.4 | 47.1 | 46 | 46.2 | 45.9 | 47.1 | 46.8 | 0.5 | | |
| 44 | 33.3316 | 75.7271 | HH mg | 82 | 47 | 47.2 | 46.6 | 46.9 | 46.2 | 47.4 | 46.5 | 47.2 | 46 | 45.9 | 46.2 | 46.6 | 0.5 | | |
| 45 | 33.3443 | 75.7296 | HH mg | 92.4 | 47 | 47.2 | 48.2 | 48.1 | 47.7 | 47 | 48.2 | 46.9 | 47.3 | 48 | | 47.6 | 0.5 | | |
| 46 | 33.3578 | 75.7225 | HH mg | 93 | 50.1 | 50 | 48.8 | 48.8 | 46.4 | 48 | 47.2 | 46.9 | 47.3 | 48.2 | | 48.2 | 1.2 | | |
| 47 | 33.3383 | 75.7256 | HH mg | 94.8 | 58.3 | 56.9 | 55 | 57.2 | 52.6 | 55.6 | 55.9 | 56 | 56.3 | 57.2 | 55.2 | 56.0 | 1.4 | | |
| 48 | 33.3497 | 75.7329 | HH mg | 100.1 | 60.5 | 58.9 | 60.1 | 59.2 | 59 | 60 | 58.3 | 58.8 | 58.9 | 59 | 59.1 | 59.3 | 0.6 | | |
| 49 | 33.3227 | 75.743 | HH mg | 102.5 | 62.9 | 60.7 | 61 | 61.5 | 60.3 | 60.5 | 59.9 | 60.2 | 60.3 | 59.9 | 62 | 60.8 | 0.9 | | |
| 50 | 33.3577 | 75.7345 | HH mg | 104.2 | 50.6 | 54.7 | 53.9 | 53.8 | 54 | 50.8 | 52 | 53.9 | 54 | 52.5 | 53.2 | 53.0 | 1.3 | | |
| 51 | 33.3603 | 75.749 | HH mg | 105.8 | 56.2 | 56 | 55.9 | 54.3 | 51.9 | 56.8 | 57 | 55.5 | 56 | 53.8 | 55.7 | 55.4 | 1.4 | | |
| 52 | 33.3737 | 75.7539 | LH schist | 108.2 | 37.9 | 36.9 | 37.8 | 39.3 | 38.2 | 38 | 38.2 | 37.7 | 39.2 | 38 | | 38.1 | 0.7 | | |
| 53 | 33.377 | 75.7759 | LH qt | 110.2 | 54.9 | 53.4 | 54 | 54.4 | 52.5 | 53.2 | 55 | 54.7 | 54.1 | 55 | 54.2 | 54.1 | 0.8 | 51.6 | 4.8 |
| 54 | 33.3708 | 75.7879 | LH qt | 112.1 | 51 | 51.9 | 53.3 | 54.2 | 54.5 | 52.9 | 52.8 | 52.5 | 53.8 | 53 | | 53.0 | 1.0 | | |
| 55 | 33.3586 | 75.8062 | LH qt | 114.7 | 49.9 | 50.3 | 50.5 | 49.5 | 51 | 51 | 50.2 | 49.2 | 50.3 | 49.8 | | 50.2 | 0.6 | | |
| 56 | 33.3578 | 75.8286 | LH qt | 115.8 | 50.5 | 52.1 | 50.4 | 50.9 | 48.4 | 47.9 | 52 | 51.1 | 50.5 | 50.9 | | 50.5 | 1.3 | | |
| 57 | 33.353 | 75.8478 | LH qt | 117.6 | 48.7 | 48.9 | 50 | 49.4 | 47.3 | 47 | 47.1 | 48.4 | 48.2 | 46.5 | | 48.2 | 1.1 | | |
| 58 | 33.3535 | 75.8654 | LH qt | 121 | 50.5 | 54.1 | 55 | 53.2 | 55.9 | 52.1 | 54.8 | 54.1 | 53.1 | 55.4 | | 53.8 | 1.6 | | |
| 59 | 33.3412 | 75.8917 | LH qt | 123.2 | 57.2 | 53.1 | 53.2 | 56 | 55.7 | 52.7 | 57.1 | 54.3 | 56.6 | 56.3 | | 55.2 | 1.6 | | |
| 60 | 33.3438 | 75.9054 | LH qt | 126.5 | 57.8 | 56.9 | 57.1 | 53.3 | 59 | 55 | 57.2 | 58.8 | 57.5 | | | 57.0 | 1.7 | | |
| 61 | 33.3437 | 75.9238 | LH qt | 130.9 | 52.9 | 53 | 55.3 | 54.6 | 55 | 52.4 | 57.7 | 53.3 | 55.2 | 54.3 | | 54.4 | 1.5 | | |
| 62 | 33.3349 | 75.9373 | LH qt | 132.3 | 40.3 | 45.2 | 37.3 | 37 | 38.3 | 44 | 39.2 | 41 | 41.1 | 36.9 | | 40.0 | 2.7 | | |

| No | Lat | Long | Lith | Dist | R1 | R2 | R3 | R4 | R5 | R6 | R7 | R8 | R9 | R10 | R11 | R12 | Mean | SD | | |
|---|---|---|---|---|---|---|---|---|---|---|---|---|---|---|---|---|---|---|---|---|
| 63 | 33.3381 | 75.9574 | LH gr | 133.8 | 53.6 | 53 | 54.5 | 54.1 | 52.8 | 52.5 | 53.3 | 53.9 | 50.8 | 52.9 | | | 53.1 | 1.0 | 55.4 | 3.0 |
| 64 | 33.3272 | 75.9669 | LH gr | 135 | 49.9 | 48.8 | 49.1 | 53 | 50.7 | 48.3 | 50.8 | 53.5 | 50.6 | 52.4 | 53.1 | | 50.9 | 1.8 | | |
| 65 | 33.3183 | 75.9932 | LH gr | 138.4 | 55.8 | 55.7 | 57.1 | 57.3 | 57 | 56.7 | 56.4 | 55.9 | 56 | 56.9 | 57 | | 56.5 | 0.6 | | |
| 66 | 33.3228 | 76.0164 | LH gr | 139.6 | 56.1 | 56.8 | 57 | 57.4 | 57.3 | 58.7 | 58 | 58.1 | 57.9 | 58.5 | 58.7 | 58 | 57.7 | 0.8 | | |
| 67 | 33.3293 | 76.0406 | LH gr | 140 | 58 | 58.1 | 58.9 | 60.1 | 57.8 | 57.9 | 58 | 58.5 | 58.2 | 57.9 | | | 58.3 | 0.7 | | |
| 68 | 33.3164 | 76.0556 | LH qt | 141.6 | 62.6 | 61.1 | 60.7 | 61.1 | 60.5 | 60.2 | 59.7 | 60.5 | 61.2 | 60.7 | | | 60.8 | 0.7 | | |
| 69 | 33.3123 | 76.07 | LH qt | 143.6 | 56.6 | 56.9 | 57 | 56.4 | 56.6 | 57.5 | 55.2 | 56.8 | 56.7 | 58 | | | 56.8 | 0.7 | | |
| 70 | 33.3055 | 76.0779 | LH gr | 147.8 | 59 | 59.3 | 57.8 | 58.8 | 60.1 | 59.2 | 58.9 | 59.7 | 59 | 57.3 | | | 58.9 | 0.8 | 58.1 | 1.6 |
| 71 | 33.2967 | 76.0874 | LH gr | 148.4 | 59.9 | 59.3 | 59.8 | 57.8 | 58 | 58.6 | 59.9 | 60.5 | 59.9 | 59.8 | | | 59.4 | 0.9 | | |
| 72 | 33.2925 | 76.1009 | LH gr | 152.5 | 60.3 | 60.2 | 61.2 | 60 | 58.9 | 59.2 | 59.2 | 59.9 | 60.1 | 59.1 | | | 59.8 | 0.7 | | |
| 73 | 33.2783 | 76.1112 | LH gr | 154.6 | 58.6 | 56.9 | 56.8 | 57 | 57.5 | 58.2 | 56 | 56.6 | 55.9 | 56 | | | 57.0 | 0.9 | | |
| 74 | 33.2628 | 76.119 | LH gr | 157.6 | 59 | 56.7 | 55 | 55.4 | 55.9 | 57.2 | 57.5 | 57 | 58.3 | 55.5 | | | 56.8 | 1.2 | | |
| 75 | 33.2586 | 76.1389 | LH gr | 158.8 | 58.2 | 57.7 | 57.7 | 55.9 | 56 | 58.2 | 58.4 | 55 | 55.2 | | | | 56.9 | 1.3 | | |

Table ST1: Details of site-specific R-values collected using rock-rebound hammer. [Abbreviations: LH MS – Lesser Himalayan Metasediments,

LH gr – Lesser Himalayan granite, LH qt – Lesser Himalayan quartzite, HHCS – Higher Himalayan crystalline sequence, HH mg – Higher

Himalayan migmatites).

---

## Referee Report (RR1)

**Review of Structural variations in basal decollement and internal deformation of the Lesser Himalayan Duplex trigger landscape morphology in NW Himalayan interiors by Dey et al.**

**Reviewer: Yann Gavillot**
**October 2020**
ygavillot@mtech.edu

I am pleased to see that Dey et al. were able to address many of the comments and concerns by the other reviewers, and provided further details, justification, and graphics in support of their morphometric analysis. This revised manuscript is a significant improvement, and I believe their analyses strengthen their interpretation of structural control associated with multiple ramps along the MHT as observed from spatial distribution of knickpoints, ksn, etc. across the Kishtwar Window of the Kashmir Himalaya.

However, there are still shortcomings in how the data are being interpreted in their structural relation to the active faulting within the duplex, kinematics of duplex growth, evidence for and against out-of-sequence thrusting and shortening rates. In general, the paper is in better shape to be considered for publication, however, I strongly recommend for another round of revision before being accepted for publication until the authors make the necessary changes. It is unfortunate that my first reviews submitted to the Associate Editor on Aug 3 fell through the cracks and were not communicated to the author prior to this revision.

I am providing here an annotated pdf with my details reviews and comments based on the newest version submitted by Dey et al. on Sept 2020. Many of these changes can be addressed in the text, but there are some fundamental inconsistencies in the structural interpretation and shortening rates calculation that may require a revised approach. I am listing below my main concerns and recommendations.

1. Throughout the paper the authors are unclear or ambiguous how they characterize and interpret patterns associated with "faulting", "growth", or "active surface faulting" across the Kishtwar Window. For instance their cross section and map figures clearly indicate their interpretation of an out-of-sequence surface breaking fault, however, in the text their interpretation is unclear going back and forth between implying active faulting on MHT crustal ramps (no surface faulting), to active duplex growth, to active out-of-sequence deformation (surface faulting) that links to a crustal ramps. The authors need to clarify and revise many of the structural terms and interpretations being used in the text to streamline their interpretations.

2. The authors need to provide more justification in their interpretation that morphometric indices provide evidence for or against active out-of-sequence faulting. In many places, the authors jump to their preferred model but failed to recognize that these morphometric indices or other structural data are non-unique! In other words, the authors do not provide enough justification why or why not the pattern observed can be attributed to a specific deformation pattern. The authors have a tendency to have model-driven interpretations and do not justify why other structural models or non-tectonic controls are not permissible. For instance, many of the arguments used by the authors as "evidence" of an active out-of-sequence fault within the KW are not justified and highly speculative. Many

of the observation described by the authors can be equally or more easily explained of the presence of an exhumed duplex floor thrust, and all of the knickpoints pattern are controlled by translation across MHT ramp which require no surface faulting.

3. I believe there would be strong benefits in this paper to recognize that this an open-ended interpretation (active out-of-sequence thrusting versus translation across MHT ramp). A more constructive approach would be to offer two possible viable structural interpretation of the cross section diagrams (out-of-sequence thrust versus and exhumed duplex floor thrust with active deeper ramp along the MHT), and let the reader see how each models can explain some of the observations.

4. Interpretation of duplex or cross section needs revision. There are several keys issues with the cross section listed below:

   a. The figure shows an out-of-sequence thrust in the KW core that projects to cut roof thrust. Because the author interprets as an out-of-sequence thrust, it implies it does not join the roof thrust as duplex. Hence this is not consistent with the discussion in the text that there is active/ongoing duplex growth. The diagram imply growth within KW occurs via out-of-sequence thrusting (ramp #1), and translation of above MHT ramp (ramp #2). There is no active/ongoing duplex kinematics as shown in the diagram.
   b. This cross section does not appear to be restorable. Even If one would attempt to retro-deform this cross section, it would show no duplex nor significant crustal thickening, but instead a single nappe in the LHS that has been faulted by an out-of-sequence fault.
   c. This cross section has major implications at odds with constraints from regional shortening absorbed in the Kashmir Himalaya orogenic wedge. Available long-term kinematics from low-temperature thermochron, Pleistocene-Holocene shortening rates, and geodetic shortening rates across the Kashmir Sub-Himalaya imply that no significant surface faulting within the KW or High Himalaya is needed to account for the total budget of plate convergence absorbed in the Kashmir Himalaya.
   d. The cross-section diagram is not consistent with duplex kinematics. Instead this pattern is more aligned with antiformal dome with flexural flow within the structure (~local crustal extrusion model).

5. Calculations and analyses of the shortening rates need substantive improvements. The author makes incorrect assumptions that incision rate deduced west of KT and KW can be translated to a shortening rate on the MCR1 fault ramp within the KW. For instance, if looking at Fig. 8b, slip on the MCR-2 would not be translated with the same geometry to the surface, as the underlying ramp-flat geometry would predict very little rock uplift translated to the surface. I would recommend the author deletes text on the shortening rate, because there is no data of incision rates in the upper plate of the inferred out-of-sequence thrust to justify a calculation of shortening rate, and it is actually not relevant to the main points of the paper. This section appears out of place for this paper.

6. Much of the text and analyses on the OSL ages appear rushed and needs revision. Details of OSL lab methodology ought to be placed in the supplement.

7. There are still many unclear sentences that need revision. At times, there are also odd choices of words and excessive use of unnecessary adjectives.

[revised manuscript text omitted]

---

## Editor Decision (ED1)

[revised manuscript text omitted]

Figure 2

[Figure]

Figure 3

[Figure]

                          Figure 4

[Figure]

                        Figure 5

[Figure]

Figure 6

[Figure]

Figure 7

[Figure]

**Figure 8**

[Figure]

[Figure]

[Figure]

[Figure]

Figure 9

[Figure]

Figure 10

[Figure]

Table 1

| Parameter | downstream | KZ1 | % change | ratio KZ1:downstream | downstream | KZ2 | % change | ratio KZ2:downstream |
|---|---|---|---|---|---|---|---|---|
| average channel gradient (m/m) | 0.006 | 0.021 | 250 | 3.5 | 0.01 | 0.046 | 360 | 4.6 |
| average channel width (m) | 70 | 45 | -35.71 | 0.6 | 55 | 42 | -24 | 0.76 |
| *Specific stream power (SSP) | 0.000086 | 0.000467 | 444.44 | 5.4 | 0.000182 | 0.001095 | 502 | 6.02 |

\* SSP calculated by assuming equal-discharge (Q)

Table 2

| Sample type | Sample name | Lat (°) | Long (°) | U (ppm) | Th (ppm) | K (%) | water (%) | Dose rate (Gy/ky) | De (Gy) | OD (%) | Age (ky) | fading correction | Corrected age (ky) |
|---|---|---|---|---|---|---|---|---|---|---|---|---|---|
| **using central age model** | | | | | | | | | | | | | |
| OSL | K02 | 33.29607 | 75.77619 | 3.8 | 7.2 | 0.46 | 6.1 | 1.74±0.02 | 141±8 | 19.5 | 81.1±4.6 | | |
| OSL | K11 | 33.35352 | 75.74649 | 3.1 | 12.7 | 2.41 | 6 | 3.97±0.09 | 341±19 | 16.8 | 85.7±5.1 | | |
| OSL | K01 | 33.15222 | 75.66323 | 2.9 | 13.2 | 2.03 | 9 | 3.88±0.04 | 193±11 | 22.1 | 49.8±2.9 | | |
| OSL | K06 | 33.15243 | 75.70609 | 3.4 | 18 | 2.17 | 5.4 | 3.97±0.05 | 205±10 | 14.4 | 51.6±2.4 | | |
| IRSL | K07 | 33.2778 | 75.76922 | 3.3 | 13.8 | 2.31 | 5.3 | 4.67±0.22 | 489±29 | 16.8 | 104.5±5.9 | 0.89 | 113±6 |
| IRSL | K08 | 33.2778 | 75.76922 | 3.5 | 16.9 | 1.97 | 5.6 | 4.61±0.23 | 528±38 | 20.5 | 114.4±6.3 | | |
| IRSL | K09 | 33.2778 | 75.76922 | 3.3 | 12.2 | 1.98 | 4.8 | 4.29±0.20 | 510±42 | 18.1 | 119.2±6.8 | 1.11 | 132±7 |
| **using minimum age model** | | | | | | | | | | | | | |
| OSL | K16 | 33.34873 | 75.73324 | 3.5 | 16.8 | 2.03 | 7.5 | 3.95±0.1 | 90±8 | 40 | 22.8±2.1 | | |
| OSL | K17 | 33.34873 | 75.73324 | 3.4 | 18 | 2.17 | 10.5 | 3.96±0.11 | 81±3.5 | 46 | 20.5±1.0 | | |
| **saturated sample** | | | | | | | | | | | | | |
| OSL | K18 | 33.35176 | 75.74325 | 3.3 | 18.7 | 2.61 | 4.5 | 4.36±0.13 | 227±14 | | 52.1±2.8 | | |

---

## Author Response (AR2)

**Manuscript # esurf-2020-37**

Answers to review comments (Please note that the annotated manuscript file is attached separately)

Dear editor and reviewers,

First of all, I would like to thank all of you for such an in-depth review for our manuscript. We really benefitted from your comments and suggestions. On behalf of the authors, I am happy to tell you that we have made significant changes in the manuscript according to your comments; especially we have looked more in detail in the geological background and discussion section. This has increased the length of the manuscript by a bit, but I ensure that all your queries are hopefully taken care of.

A few key changesa. Title has been revised.
b. Abstract has been revised and reduced in length.
c. End of introduction has been modified.
d. Geological background section: debate on structural setting of the Kishtwar Window elaborated.
e. Methods: Luminescence sample process protocol is now moved to the appendix.
f. Results: Couple more OSL data included.
g. Discussion is revised to offer an open-ended view on the debate between duplex-growth vs. active out-of-sequence faulting in the interior of the Kashmir Himalaya.
h. Conclusions revised accordingly.

Thank you again for considering our work.

On behalf of the authors

Saptarshi Dey

**Comments to reviewer #1**

1. General query regarding citations and figure references

   Revised and rechecked twice.

2. Syntax errors and nomenclature issues.

   Revised.

3. The geomorphic data still seem to be not fully consistent. Let us look, e.g., at the channel slope and the steepness index ksn in Fig. 6. They are related to each other by the factor A^0.9 where A is the catchment size which increases downstream, i.e., from the right to the left. Let us start from the right-hand edge where the slope increases while ksn immediately decreases to the left. Or look how much the large double peak at x = 10 km is higher than the ksn values at x < 10 km, and how much lower the relative difference in slope is. This is in principle impossible. I am quite sure that this is not a fundamental problem, maybe even only inconsistent smoothing of the data. I think it can be fixed easily, but such things just do not make it easier to trust that everything is technically sound -- although the most importing parts are probably.

   Agreed and revised. You are right that there is some issue with smoothing. Probably the preprocessing of the DEM data has some issues. Now we have used a 10 point smoothing window throughout the entire stretch.

**Comments to reviewer #2**

1. Throughout the paper the authors are unclear or ambiguous how they characterize and interpret patterns associated with "faulting", "growth", or "active surface faulting" across the Kishtwar Window. For instance their cross section and map figures clearly indicate their interpretation of an out-of-sequence surface breaking fault, however, in the text their interpretation is unclear going back and forth between implying active faulting on MHT crustal ramps (no surface faulting), to active duplex growth, to active out-of-sequence deformation (surface faulting) that links to a crustal ramps. The authors need to clarify and revise many of the structural terms and interpretations being used in the text to
streamline their interpretations.

Text has been revised. Now, we invite for an open interpretation of our results. We show
that our results can be explained by both duplex-growth model aided by differential uplift
over mid-crustal ramps or by active out-of-sequence faulting across the window.
Although our field observation questions the brittle-deforming duplex model, we neither
have any first-hand evidence for regional out-of-sequence faulting too.

2.  The authors need to provide more justification in their interpretation that morphometric
indices provide evidence for or against active out-of-sequence faulting. In many places,
the authors jump to their preferred model but failed to recognize that these morphometric
indices or other structural data are non-unique! In other words, the authors do not provide
enough justification why or why not the pattern observed can be attributed to a specific
deformation pattern. The authors have a tendency to have model-driven interpretations
and do not justify why other structural models or non-tectonic controls are not
permissible. For instance, many of the arguments used by the authors as "evidence" of an
active out-of-sequence fault within the KW are not justified and highly speculative. Many
of the observation described by the authors can be equally or more easily explained of the
presence of an exhumed duplex floor thrust, and all of the knickpoints pattern are
controlled by translation across MHT ramp which require no surface faulting.

Revised in order to keep the interpretation open.

3.  I believe there would be strong benefits in this paper to recognize that this an open-ended
interpretation (active out-of-sequence thrusting versus translation across MHT ramp). A
more constructive approach would be to offer two possible viable structural interpretation
of the cross section diagrams (out-of-sequence thrust versus and exhumed duplex floor
thrust with active deeper ramp along the MHT), and let the reader see how each models
can explain some of the observations.

Agreed and revised accordingly. Please refer to the end of discussion section (line #655-
696).

4.  Interpretation of duplex or cross section needs revision. There are several keys issues
with the cross section listed below:

5.  The figure shows an out-of-sequence thrust in the KW core that projects to cut roof thrust. Because the author interprets as an out-of-sequence thrust, it implies it does not join the roof thrust as duplex. Hence this is not consistent with the discussion in the text that there is active/ongoing duplex growth. The diagram imply growth within KW occurs via out-of-sequence thrusting (ramp #1), and translation of above MHT ramp (ramp #2). There is no active/ongoing duplex kinematics as shown in the diagram.

    Cross-section is revised in Fig.8d. We compare between 'duplex-growth by slip on MHT' model vs. 'active out-of-sequence faulting' models in Fig.8d. Both the models can explain the observed morphometric variations.

6.  This cross section does not appear to be restorable. Even If one would attempt to retro-deform this cross section, it would show no duplex nor significant crustal thickening, but instead a single nappe in the LHS that has been faulted by an out-of-sequence fault.

    Our field observation tells that the Chail nappe exposed in the KW is internally buckle-folded, pervasive folding and flexure has resulted into crustal thickening (please refer to our field photos in Fig.2). This is supported by Fuchs (1975), Frank et al., (1995).

7.  This cross section has major implications at odds with constraints from regional shortening absorbed in the Kashmir Himalaya orogenic wedge. Available long-term kinematics from low-temperature thermochron, Pleistocene-Holocene shortening rates, and geodetic shortening rates across the Kashmir Sub-Himalaya imply that no significant surface faulting within the KW or High Himalaya is needed to account for the total budget of plate convergence absorbed in the Kashmir Himalaya.

    Agreed. We see that over Quaternary and geodetic timescales, the total Himalayan shortening is steady 13-15 mm/yr. Published shortening rates accommodated in Sub-Himalaya over millennial timescales also hint similar amount of shortening rates. That leaves no shortening to be accommodated beyond the MBT. But, this has a major assumption that the total shortening rate since late Pleistocene is equal to other timescales. This assumption has been questioned by the work of Vassallo et al., (2015). In line with this, we do not pinpoint that 'it has to be an out-of-sequence faulting in the KW', but 'it could be an out-of-sequence faulting in the KW'.

8. d. The cross-section diagram is not consistent with duplex kinematics. Instead this pattern is more aligned with antiformal dome with flexural flow within the structure (~local crustal extrusion model).

The cross-section diagram has now been removed and revised. However, we like to comment that the cross-section is a rough representation of field data and our field observation goes against the existence of multiple nappes forming the duplex. In lieu of that, we find tightly-folded Chail nappe in the window. Our observation is close to what is proposed by Fuchs (1975) and Frank et al., (1995). In revision, we added the comparison of two models – duplex-growth model (Gavillot et al., 2018) and out-of-sequence model (Fig.8d). This keeps the manuscript open for the readers to decide which one they favor.

9. Calculations and analyses of the shortening rates need substantive improvements. The author makes incorrect assumptions that incision rate deduced west of KT and KW can be translated to a shortening rate on the MCR1 fault ramp within the KW. For instance, if looking at Fig. 8b, slip on the MCR-2 would not be translated with the same geometry to the surface, as the underlying ramp-flat geometry would predict very little rock uplift translated to the surface. I would recommend the author deletes text on the shortening rate, because there is no data of incision rates in the upper plate of the inferred out-of-sequence thrust to justify a calculation of shortening rate, and it is actually not relevant to the main points of the paper. This section appears out of place for this paper.

We have deleted the discussion on shortening rates.

10. Much of the text and analyses on the OSL ages appear rushed and needs revision. Details of OSL lab methodology ought to be placed in the supplement.

OSL methodology has been revised (line #337-356). Texts are added in justification of luminescence ages. Additional information regarding OSL measurements given in Supplementary figure B5.

11. There are still many unclear sentences that need revision. At times, there are also odd choices of words and excessive use of unnecessary adjectives.

Sentences and 'odd' words have been revised thoroughly.

12. In Fig.6 and 7, knickpoint D1 and L2 not visible.

This is due to the scale of the graphic. The knickpoints are smaller (dz value: ~30m) while the vertical scale is ~1cm = 500m/1000m and horizontal scale is 1cm = 20 km. However, if you take a look in IFg.7d, you will observe slight increase in parameter value. Said so, L2 is far downstream from our area of interest.

** All the comments found in the annotated referee report is answered or addressed in the revised version.

[revised manuscript text omitted]

Figure 2

[Figure]

Figure 3

[Figure]

                          Figure 4

[Figure]

Figure 5

[Figure]

Figure 6

[Figure]

Figure 7

[Figure]

Figure 8

[Figure]

[Figure]

[Figure]

[Figure]

Figure 9

[Figure]

Figure 10

[Figure]

Table 1

| Parameter | flat 1 | ramp 1 | % change | ratio ramp 1:flat 1 | flat 2 | ramp 2 | % change | ratio ramp 2:flat 2 |
|---|---|---|---|---|---|---|---|---|
| average channel gradient (m/m) | 0.006 | 0.021 | 250.00 | 3.5 | 0.01 | 0.046 | 360 | 4.60 |
| average channel width (m) | 70 | 45 | -35.71 | 0.6 | 55 | 42 | -24 | 0.76 |
| *Specific stream power (SSP) | 0.000086 | 0.000467 | 444.44 | 5.4 | 0.000182 | 0.001095 | 502 | 6.02 |

* SSP calculated by assuming equal-discharge (Q)

Table 2

| Sample type | Sample name | Lat (°) | Long (°) | U (ppm) | Th (ppm) | K (%) | water (%) | Dose rate (Gy/ky) | De (Gy) | OD (%) | Age (ky) | fading correction | Corrected age (ky) |
|---|---|---|---|---|---|---|---|---|---|---|---|---|---|
| **using central age model** | | | | | | | | | | | | | |
| OSL | K02 | 33.29607 | 75.77619 | 3.8 | 7.2 | 0.46 | 6.1 | 1.74±0.02 | 141±8 | 19.5 | 81.1±4.6 | | |
| OSL | K11 | 33.35352 | 75.74649 | 3.1 | 12.7 | 2.41 | 6 | 3.97±0.09 | 341±19 | 16.8 | 85.7±5.1 | | |
| OSL | K01 | 33.15222 | 75.66323 | 2.9 | 13.2 | 2.03 | 9 | 3.88±0.04 | 193±11 | 22.1 | 49.8±2.9 | | |
| OSL | K06 | 33.15243 | 75.70609 | 3.4 | 18.0 | 2.17 | 5.4 | 3.97±0.05 | 205±10 | 14.4 | 51.6±2.4 | | |
| IRSL | K07 | 33.27780 | 75.76922 | 3.3 | 13.8 | 2.31 | 5.3 | 4.67±0.22 | 489±29 | 16.8 | 104.5±5.9 | 0.89 | 113±6 |
| IRSL | K08 | 33.27780 | 75.76922 | 3.5 | 16.9 | 1.97 | 5.6 | 4.61±0.23 | 528±38 | 20.5 | 114.4±6.3 | | |
| IRSL | K09 | 33.27780 | 75.76922 | 3.3 | 12.2 | 1.98 | 4.8 | 4.29±0.20 | 510±42 | 18.1 | 119.2±6.8 | 1.11 | 132±7 |
| **using minimum age model** | | | | | | | | | | | | | |
| OSL | K16 | 33.34873 | 75.73324 | 3.5 | 16.8 | 2.03 | 7.5 | 3.95±0.1 | 90±8 | 40 | 22.8±2.1 | | |
| OSL | K17 | 33.34873 | 75.73324 | 3.4 | 18 | 2.17 | 10.5 | 3.96±0.11 | 81±3.5 | 46 | 20.5±1.0 | | |
| **saturated sample** | | | | | | | | | | | | | |
| OSL | K18 | 33.35176 | 75.74325 | 3.3 | 18.7 | 2.61 | 4.5 | 4.36±0.13 | 227±14 | | 52.1±2.8 | | |

---

## Author Response (AR3)

**Review comments and answers**

**ESD manuscript (esurf-2020-37)**

In their revised manuscript, *'Implications of the ongoing rock uplift in NW Himalayan interiors'* Dey et al., combine topographic analyses with a variety of field observations, dating of Quaternary deposits, and synthesis of previous work to explore the extent to which the surface geology and geomorphology helps to constrain the subsurface structural geometry in the northwestern Himalaya. The paper is improved from the original submission and the authors have addressed some of the questions I had on the first draft. There are still some issues with following the arguments, though mostly all the information is there, there just needs to be a little more thought given to organization or explanation as to where the authors are taking the readers. In a few places the authors need to provide a bit more explanation that was asked for in the previous review as well (e.g., the specific stream power calculation, though ultimately these values are barely used in the paper, so the details might not matter that much).

Thanks for your thorough suggestions in the first round and in the second round, too. We acknowledge that the discussion has been a bit wayward than we could/should have done before. We spent time on how to improve the discussion and revised the discussion section considerably. For example, now we evaluate our observation and results in light of two competing models we refer: duplex-growth model and out-of-sequence fault-ramp model. At present, our data show similarity with long-term exhumation rates, but. Honestly it cannot resolve which model is perfect for the setting. This is because we have field observations/ results which partially agree to both the models. And, that's why we didn't favour any model in the end, but, tried to keep it open. We have explained the revisions/ changes made in the following paragraphs.

1. This comment is mainly on the discussion as a whole, but it kind of extends to the results as well. At present, it's kind of hard to follow the connections between the various data being presented, i.e. what is the relevance of the different parts to each other. You go through some of this in the very

beginning of the paper, but by the time a reader gets to the results and discussion, it gets harder to see these connections. You have the start of a description of what you're going to do near the beginning of your discussion (L483-491), but I think this could benefit from a little expansion and a bit more specificity before you launch into the rest of the discussion.

We revised the starting paragraphs of the discussion section and made it more concise (line #470-488), so that readers can follow our arguments provided in sub-sections 5.1-5.5.

2. Where every thing comes together is Section 5.5, but as highlighted in some of my line by line comments below, this section is hard to follow and it's not clear what model you prefer or if you prefer any of them. It seems like you are arguing against the Gallivot duplex model at the beginning, but then you agree with parts of it. In the end, it seems like the data you have assembled can't discriminate between some of these options or clarify some of the details. This is fine, but you just need to be much more explicit about it, i.e. explain from the beginning that you can't differentiate on the basis of the data you have and then go through the relevant data and how it does or does not support various models. You tell us this is the case in the introduction, but this final discussion section meanders around this point a bit.

Section 5.5 is now split into two sub-sections 5.4 Our new results in context with the previously-published data and 5.5 Two competing models: duplex-growth model vs. out-of-sequence fault-ramp model. Section 5.4 deals with the comparison of our data with the data presented by Gavillot et al., (2018) from Kishtwar window and data from Dhauladhar Range (Thiede et al., 2017), Kullu-Rampur Window (Stuebner et al., 2018). Section 5.5 contains the competency of the two models with our results and field observations. We hope that this should make the manuscript/ discussion to be pointier.

Alongside this, we have removed some parts of the discussion, which we believe is either repetition of earlier statements or are speculative at this moment as those don't have independent data to support the claims. So, the revised text is more concise.

3. Knickpoints vs knickzones: There is some lack of consistency between defining something as knickpoint vs a knickzone which leads to confusion. Both K1 and K2 seem to both be knickpoints and knickzones. In figure 7, they are both (i.e., there are steepened zones highlighted named K1 and K2 and also discrete point named K1 and K2). In the text, K1 is described more as knickzone, but K2 is described as a knickpoint and a knickzone in section 5.1.2. Are the points for K1 and K2 just the upstream beginning of the knickzones? If that is the case, maybe distinguishing them with different names (e.g. K1_s, K2_s or something). It makes it challenging to follow how we are suppose to interpret these features.

Now in revised manuscript, knickzones are marked as KZ and knickpoints (discrete ones and the upstream head of the knickzones) are marked as KP. This will clear the confusion. All the occurrences including figures are revised.

Line by line:

L273-283: Could you provide a little more detail on this calculation, kind of just fill in the gaps. Specifically, I assume you use the width measurements from the previous section in your calculation of SSP? How are you estimating discharge? Again, I would assume you're routing the TRMM rainfall, but saying this is necessary. Do you assume a runoff ratio of 1? Do you have any data that would allow you to estimate if a constant runoff ratio is warranted (regardless of the specific value), etc?

Revised and new text added at line#279-283. In short, as we see TRMM data, we see no variation in rainfall across the study area, that is why we assume a constant Q. And, at this moment, we don't have any data on run-off ratio, so we assume that Q translates completely into run-off.

L392-402: Some of this text is repeated from your background section. Maybe keeping the previous observations in the background and your new observations here in the results would be better.

Text revised accordingly.

L652-653: It's not clear which field observations don't work with the duplex model or what the differences would be. Later (L658) you mention that the Gavillot model suggest no surface-faulting, but in the sentence that precedes this, you describe that the observed deformation could be a product of active faulting, crustal buckling, or internal folding, so the incompatibility of these observations with this model is not clear.

Discussion is revised and a new sub-section is added (line # 650-686) titled 'Two competing models: duplex-growth model vs. out-of-sequence fault ramp model'. In this section we categorize our observations and their match and mismatch with either models.

L669-672: This seems out of place and more like something that belongs in your conclusion.

Revised and deleted from the text. The conclusion is also revised accordingly.

---

## Author Response (AR4)

Dear Editor,

In the latest version, all recommended corrections have been accepted. A few of typographical mistakes have been revised. Tables are revised in order to remove colored panels. Figures and fonts have been checked for visibility and consistency with text.

Thank you for your help in the whole process.

On behalf of the authors,

Saptarshi